# Manifold Mixup: Learning Better Representations by Interpolating Hidden States

## Abstract

Deep networks often perform well on the data distribution on which they are trained, yet give incorrect (and often very confident) answers when evaluated on points from off of the training distribution. This is exemplified by the adversarial examples phenomenon but can also be seen in terms of model generalization and domain shift. Ideally, a model would assign lower confidence to points unlike those from the training distribution. We propose a regularizer which addresses this issue by training with interpolated hidden states and encouraging the classifier to be less confident at these points. Because the hidden states are learned, this has an important effect of encouraging the hidden states for a class to be concentrated in such a way so that interpolations within the same class or between two different classes do not intersect with the real data points from other classes. This has a major advantage in that it avoids the underfitting which can result from interpolating in the input space. We prove that the exact condition for this problem of underfitting to be avoided by Manifold Mixup is that the dimensionality of the hidden states exceeds the number of classes, which is often the case in practice. Additionally, this concentration can be seen as making the features in earlier layers more discriminative. We show that despite requiring no significant additional computation, Manifold Mixup achieves large improvements over strong baselines in supervised learning, robustness to single-step adversarial attacks, semi-supervised learning, and Negative Log-Likelihood on held out samples.

## 1 Introduction

Machine learning systems have been enormously successful in domains such as vision, speech, and language and are now widely used both in research and industry. Modern machine learning systems typically only perform well when evaluated on the same distribution that they were trained on. However machine learning systems are increasingly being deployed in settings where the environment is noisy, subject to domain shifts, or even adversarial attacks. In many cases, deep neural networks which perform extremely well when evaluated on points on the data manifold give incorrect answers when evaluated on points off the training distribution, and with strikingly high confidence.

This manifests itself in several failure cases for deep learning. One is the problem of adversarial examples (Szegedy et al., 2014), in which deep neural networks with nearly perfect test accuracy can produce incorrect classifications with very high confidence when evaluated on data points with small (imperceptible to human vision) adversarial perturbations. These adversarial examples could present serious security risks for machine learning systems. Another failure case involves the training and testing distributions differing significantly. With deep neural networks, this can often result in dramatically reduced performance.

To address these problems, our *Manifold Mixup* approach builds on following assumptions and motivations: (1) we adopt the manifold hypothesis, that is, data is concentrated near a lower-dimensional non-linear manifold (this is the only required assumption on the data generating distribution for *Manifold Mixup* to work); (2) a neural net can learn to transform the data non-linearly so that the transformed data distribution now lies on a nearly flat manifold; (3) as a consequence, linear interpolations between examples in the hidden space also correspond to valid data points, thus providing novel training examples.

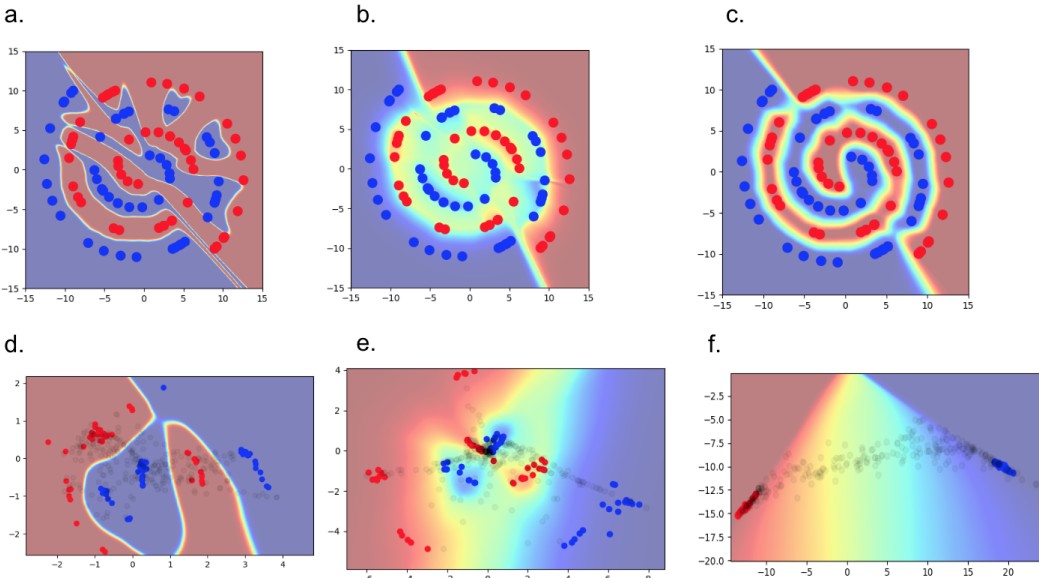

Figure 1: The top row (a,b,c) shows the decision boundary on the 2d spirals dataset trained with a baseline model (a fully connected neural network with nine layers where middle layer is a 2D bottleneck layer), Input Mixup with $\alpha = 1.0$, and *Manifold Mixup* applied only to the 2D bottleneck layer. As seen in (b), Input Mixup can suffer from underfitting since the interpolations between two samples may intersect with a real sample. Whereas *Manifold Mixup* (c), fits the training data perfectly (more intuitive example of how *Manifold Mixup* avoids underfitting is given in Appendix H). The bottom row (d,e,f) shows the hidden states for the baseline, Input Mixup, and manifold mixup respectively. *Manifold Mixup* concentrates the labeled points from each class to a very tight region, as predicted by our theory (Section 3) and assigns lower confidence classifications to broad regions in the hidden space. The black points in the bottom row are the hidden states of the points sampled uniformly in x-space and it can be seen that manifold mixup does a better job of giving low confidence to these points. Additional results in Figure 6 of Appendix B show that the way *Manifold Mixup* changes the representations is not accomplished by other well-studied regularizers (weight decay, dropout, batch normalization, and adding noise to the hidden states).

*Manifold Mixup* performs training on the convex combinations of the hidden state representations of data samples. Previous work, including the study of analogies through word embeddings (e.g. king - man + woman ≈ queen), has shown that such linear interpolation between hidden states is an effective way of combining factors (Mikolov et al., 2013). Combining such factors in the higher level representations has the advantage that it is typically lower dimensional, so a simple procedure like linear interpolation between pairs of data points explores more of the space and with more of the points having meaningful semantics. When we combine the hidden representations of training examples, we also perform the same linear interpolation in the labels (seen as one-hot vectors or categorical distributions), producing new soft targets for the mixed examples.

In practice, deep networks often learn representations such that there are few strong constraints on how the states can be distributed in the hidden space, because of which the states can be widely distributed through the space, (as seen in Figure 1d). As well as, nearly all points in hidden space correspond to high confidence classifications even if they correspond to off-the-training distribution samples (seen as black points in Figure 1d). In contrast, the consequence of our *Manifold Mixup* approach is that the hidden states from real examples of a particular class are concentrated in local regions and the majority of the hidden space corresponds to lower confidence classifications. This concentration of the hidden states of the examples of a particular class into a local regions enables learning more discriminative features. A low-dimensional example of this can be seen in Figure 1 and a more detailed analytical discussion for what "concentrating into local regions" means is in Section 3.

Our method provides the following contributions:

- The introduction of a novel regularizer which outperforms competitive alternatives such as Cutout (Devries & Taylor, 2017), Mixup (Zhang et al., 2018), AdaMix (Guo et al., 2016), and Dropout (Hinton et al., 2012). On CIFAR-10, this includes a 50% reduction in test Negative Log-Likelihood (NLL) from 0.1945 to 0.0957.

- *Manifold Mixup* achieves significant robustness to single step adversarial attacks.

- A new method for semi-supervised learning which uses a *Manifold Mixup* based consistency loss. This method reduces error relative to Virtual Adversarial Training (VAT) (Miyato et al., 2018a) by 21.86% on CIFAR-10, and unlike VAT does not involve any additional significant computation.

- An analysis of *Manifold Mixup* and exact sufficient conditions for *Manifold Mixup* to achieve consistent interpolations. Unlike Input Mixup, this doesn't require strong assumptions about the data distribution (see the failure case of Input Mixup in Figure 1): only that the number of hidden units exceeds the number of classes, which is easily satisfied in many applications.

## 2 MANIFOLD MIXUP

The *Manifold Mixup* algorithm consists of selecting a random layer (from a set of eligible layers including the input layer) $k$. We then process the batch without any mixup until reaching that layer, and we perform mixup at that hidden layer, and then continue processing the network starting from the mixed hidden state, changing the target vector according to the mixup interpolation. More formally, we can redefine our neural network function $y = f(x)$ in terms of $k$: $f(x) = f_k(g_k(x))$. Here $g_k$ is a function which runs a neural network from the input hidden state $k$ to the output $y$, and $h_k$ is a function which computes the $k$-th hidden layer activation from the input $x$.

For the linear interpolation between factors, we define a variable $\lambda$ and we sample from $p(\lambda)$. Following (Zhang et al., 2018), we always use a beta distribution $p(\lambda) = Beta(\alpha, \alpha)$. With $\alpha = 1.0$, this is equivalent to sampling from $U(0, 1)$.

We consider interpolation in the set of layers $S_k$ and minimize the expected *Manifold Mixup* loss.

$$L = \mathbb{E}_{(x_i, y_i), (x_j, y_j) \sim p(x,y), \lambda \sim p(\lambda), k \sim S_k} \ell(f_k(\lambda g_k(x_i) + (1 - \lambda)g_k(x_j))), \lambda y_i + (1 - \lambda)y_j) \quad (1)$$

We backpropagate gradients through the entire computational graph, including to layers before the mixup process is applied (Section 5.1 and appendix Section B explore this issue directly). In the case where $k = 0$ is the input layer and $S_k = 0$, *Manifold Mixup* reduces to the mixup algorithm of Zhang et al. (2018). With $\alpha = 2.0$, about 5% of the time $\lambda$ is within 5% of 0 or 1, which essentially means that an ordinary example is presented. In the more general case, we can optimize the expectation in the *Manifold Mixup* objective by sampling a different layer to perform mixup in on each update. We could also select a new random layer as well as a new lambda for each example in the minibatch. In theory this should reduce the variance in the updates introduced by these random variables. However in practice we found that this didn't have a significant effect on the results, so we decided to sample a single lambda and a randomly chosen layer per minibatch.

In comparison to Input Mixup, the results in the Figure 2 demonstrate that *Manifold Mixup* reduces the loss calculated along hidden interpolations significantly better than Input Mixup, without significantly changing the loss calculated along visible space interpolations.

## 3 HOW MANIFOLD MIXUP CHANGES REPRESENTATIONS

Our goal is to show that if one does mixup in a sufficiently deep hidden layer in a deep network, then a mixup loss of zero can be achieved so long the dimensionality of that hidden layer $\dim(\mathcal{H})$ is greater than the number of classes $d$. More specifically the resulting representations for that class must fall onto a subspace of dimension $\dim(\mathcal{H}) - d$.

Assume $\mathcal{X}$ and $\mathcal{H}$ to denote the input and representation spaces, respectively. We denote the labelset by $\mathcal{Y}$ and let $\mathcal{Z} \triangleq \mathcal{X} \times \mathcal{Y}$. Also, let us denote the set of all probability measures on $\mathcal{Z}$ by $M(\mathcal{Z})$. Assume $\mathcal{G} \subseteq \mathcal{H}^{\mathcal{X}}$ to be the set of all possible functions that can be generated by the neural network

mapping input to the representation space. In this regard, each $g \in \mathcal{G}$ represents a mapping from input to the representation units. A similar definition can be made for $\mathcal{F} \subseteq \mathcal{Y}^{\mathcal{H}}$, as the space of all possible functions from the representation space to the output.

We are interested in the solution of the following problem, at least in some specific asymptotic regimes:

$$J\left(L, P\right) \triangleq \inf_{g \in \mathcal{G}, \, f \in \mathcal{F}} \mathbb{E}_\lambda \left\{ \int_{\mathcal{Z}^2} L\left(f \circ \mathrm{Mix}_\lambda\left(g\left(\boldsymbol{X}_1\right), g\left(\boldsymbol{X}_2\right)\right), \mathrm{Mix}_\lambda\left(\boldsymbol{y}_1, \boldsymbol{y}_2\right)\right) \prod_{i=1}^{2} \mathrm{d}P\left(\boldsymbol{X}_i, y_i\right) \right\},$$
(2)

where

$$\mathrm{Mix}_\lambda\left(a, b\right) \triangleq \lambda a + \left(1 - \lambda\right) b, \quad \lambda \in \left[0, 1\right],$$
(3)

for any $a$ and $b$ defined on the same domain.

We analyze the above-mentioned minimization when the probability measure $P = \mathbb{P}_{\boldsymbol{D}}$ is chosen as the empirical distribution over a finite dataset of size $n$, denoted by $\boldsymbol{D} = \left\{\left(\boldsymbol{X}_i, \boldsymbol{y}_i\right)\right\}_{i=1}^{n}$. Let $f^* \in \mathcal{F}$ and $g^* \in \mathcal{G}$ be the minimizers in (2) with $P = \mathbb{P}_{\boldsymbol{D}}$.

In particular, we are interested in the case where $\mathcal{G} = \mathcal{H}^{\mathcal{X}}$, $\mathcal{F} = \mathcal{Y}^{\mathcal{H}}$, and $\mathcal{H}$ is a vector space; These conditions simply state that the two respective neural networks which map input into representation space, and representation space to the output are being extended asymptotically[1]. In this regard, we show that the minimizer $f^*$ is a linear function from $\mathcal{H}$ to $\mathcal{Y}$. This way, it is easy to show that the following equality holds:

$$J\left(L, \mathbb{P}_{\boldsymbol{D}}\right) = \inf_{\boldsymbol{h}_1, \ldots, \boldsymbol{h}_n \in \mathcal{H}} \frac{1}{n\left(n-1\right)} \sum_{\substack{i,j=1 \\ i \neq j}}^{n} \left\{ \inf_{f \in \mathcal{F}} \int_0^1 L\left(f \circ \mathrm{Mix}_\lambda\left(\boldsymbol{h}_i, \boldsymbol{h}_j\right), \mathrm{Mix}_\lambda\left(\boldsymbol{y}_i, \boldsymbol{y}_j\right)\right) \mathrm{d}\lambda \right\},$$
(4)

where $\boldsymbol{h}_i \triangleq g\left(\boldsymbol{X}_i\right)$ is the representation of $\boldsymbol{X}_i$.

**Theorem 1.** *Assume $\mathcal{H}$ to be a vector space with dimension $\dim\left(\mathcal{H}\right)$, and let $d \in \mathbb{N}$ to represent the number of distinct classes in dataset $\boldsymbol{D}$. Then, if $\dim\left(\mathcal{H}\right) \geq d - 1$, $J\left(L, \mathbb{P}_{\boldsymbol{D}}\right) = 0$ and the minimizer function $f^*$ is a linear map from $\mathcal{H}$ to $\mathbb{R}^d$.*

*Proof.* With basic linear algebra, one can confirm that the following argument is true as long as $\dim\left(\mathcal{H}\right) \geq d - 1$:

$$\exists \boldsymbol{A}, \boldsymbol{H} \in \mathbb{R}^{\dim(\mathcal{H}) \times d}, \boldsymbol{b} \in \mathbb{R}^d \quad \text{such that} \quad \boldsymbol{A}^T \boldsymbol{H} + \boldsymbol{b} \boldsymbol{1}_d^T = I_{d \times d},$$
(5)

where $I_{d \times d}$ and $\boldsymbol{1}_d$ are the $d$-dimensional identity matrix and all-one vector, respectively. In fact, $\boldsymbol{b}\boldsymbol{1}_d^T$ is a rank-one matrix, while the rank of identity matrix is $d$. Therefore, $\boldsymbol{A}^T \boldsymbol{H}$ only needs to be rank $d - 1$.

Let $f^*\left(\boldsymbol{h}\right) \triangleq \boldsymbol{A}\boldsymbol{h} + \boldsymbol{b}$, for all $\boldsymbol{h} \in \mathcal{H}$. Also, let $g^*\left(\boldsymbol{X}_i\right) = \boldsymbol{h}_{\zeta_i}$, where $\boldsymbol{h}_i$ here means the $i$th column of matrix $\boldsymbol{H}$, and $\zeta_i \in \left\{1, \ldots, d\right\}$ is the class-index of the $i$th sample. We show that such selections will make the objective in (2) equal to zero (which is the minimum possible value). More precisely, the following relations hold:

$$\frac{1}{n\left(n-1\right)} \sum_{\substack{i,j=1 \\ i \neq j}}^{n} \left\{ \int_0^1 L\left(f^* \circ \mathrm{Mix}_\lambda\left(g^*\left(\boldsymbol{X}_i\right), g^*\left(\boldsymbol{X}_j\right)\right), \mathrm{Mix}_\lambda\left(\boldsymbol{y}_i, \boldsymbol{y}_j\right)\right) \mathrm{d}\lambda \right\},$$

$$= \frac{1}{n\left(n-1\right)} \sum_{\substack{i,j=1 \\ i \neq j}}^{n} \left\{ \int_0^1 L\left(\boldsymbol{A}^T\left(\lambda \boldsymbol{h}_{\zeta_i} + \left(1-\lambda\right)\boldsymbol{h}_{\zeta_j}\right) + \boldsymbol{b}, \lambda \boldsymbol{y}_{\zeta_i} + \left(1-\lambda\right)\boldsymbol{y}_{\zeta_j}\right) \mathrm{d}\lambda \right\},$$

$$= \frac{1}{n\left(n-1\right)} \sum_{\substack{i,j=1 \\ i \neq j}}^{n} \left\{ \int_0^1 L\left(u\left(\lambda\right), u\left(\lambda\right)\right) \mathrm{d}\lambda \right\}$$

$$= 0.$$
(6)

---

[1]Due to the consistency theorem that proves neural networks with nonlinear activation functions are dense in the function space

The final equality is a direct result of $\boldsymbol{A}^T \boldsymbol{h}_{\zeta_i} + \boldsymbol{b} = \boldsymbol{y}_{\zeta_i}$ for $i = 1, \ldots, n$. □

Also, it can be shown that as long as $\dim(\mathcal{H}) > d - 1$, then data points in the representation space $\mathcal{H}$ have some degrees of freedom to move independently.

**Corollary 1.** *Consider the setting in Theorem 1, and assume* $\dim(\mathcal{H}) > d - 1$*. Let* $g^* \in \mathcal{G}$ *to be the true minimizer of* (2) *for a given dataset* $\boldsymbol{D}$*. Then, data-points in the representation space, i.e.* $g^*(\boldsymbol{X}_i)$*, fall on a* $(\dim(\mathcal{H}) - d + 1)$*-dimensional subspace.*

*Proof.* In the proof of Theorem 1, we have

$$\boldsymbol{A}^T \boldsymbol{H} = I_{d \times d} - \boldsymbol{b} \mathbf{1}_d^T. \tag{7}$$

The r.h.s. of (7) can become a rank-$(d-1)$ matrix as long as vector $\boldsymbol{b}$ is chosen properly. Thus, $\boldsymbol{A}$ is free to have a null-space of dimension $\dim(\mathcal{H}) - d + 1$. This way, one can assign $g^*(\boldsymbol{X}_i) = \boldsymbol{h}_{\zeta_i} + \boldsymbol{e}_i$, where $\boldsymbol{h}_j$ and $\zeta_i$ (for $j = 1, \ldots, d$ and $i = 1, \ldots, n$) are defined in the same way as in Theorem 1, and $\boldsymbol{e}_i$s can are arbitrary vectors in the null-space of $\boldsymbol{A}$, i.e. $\boldsymbol{e}_i \in \ker(\boldsymbol{A})$ for all $i$. □

This result implies that if the *Manifold Mixup* loss is minimized, then the representation for each class will lie on a subspace of dimension $\dim(\mathcal{H}) - d + 1$. In the most extreme case where $\dim(\mathcal{H}) = d - 1$, each hidden state from the same class will be driven to a single point, so the change in the hidden states following any direction on the class-conditional manifold will be zero. In the more general case with a larger $\dim(\mathcal{H})$, a majority of directions in $\mathcal{H}$-space will not change as we move along the class-conditional manifold.

Why are these properties desirable? First, it can be seen as a flattening [2]. of the class-conditional manifold which encourages learning effective representations earlier in the network. Second, it means that the region in hidden space occupied by data points from the true manifold has nearly zero measure. So a randomly sampled hidden state within the convex hull spanned by the data is more likely to have a classification score that is not fully confident (non-zero entropy). Thus it encourages the network to learn discriminative features in all layers of the network and to also assign low-confidence classification decisions to broad regions in the hidden space (this can be seen in Figure 1 and Figure 6).

## 4 RELATED WORK

Regularization is a major area of research in machine learning. *Manifold Mixup* closely builds on two threads of research. The first is the idea of linearly interpolating between different randomly drawn examples and similarly interpolating the labels (Zhang et al., 2018; Tokozume et al., 2018). These methods encourage the output of the entire network to change linearly between two randomly drawn training samples, which can result in underfitting. In contrast, for a particular layer at which mixing is done, *Manifold Mixup* allows lower layers to learn more concentrated features in such a way that it makes it easier for the output of the upper layers to change linearly between hidden states of two random samples, achieving better results (section 5.1 and Appendix B).

Another line of research closely related to *Manifold Mixup* involves regularizing deep networks by perturbing the hidden states of the network. These methods include dropout (Hinton et al., 2012), batch normalization (Ioffe & Szegedy, 2015), and the information bottleneck (Alemi et al., 2017). Notably Hinton et al. (2012) and Ioffe & Szegedy (2015) both demonstrated that regularizers already demonstrated to work well in the input space (salt and pepper noise and input normalization respectively) could also be adapted to improve results when applied to the hidden layers of a deep network. We believe that the regularization effect of *Manifold Mixup* would be complementary to that of these algorithms.

Zhao & Cho (2018) explored improving adversarial robustness by classifying points using a function of the nearest neighbors in a fixed feature space. This involved applying mixup between each set of nearest neighbor examples in that feature space. The similarity between Zhao & Cho (2018) and

---

[2] Please refer to Appendix I for the meaning of *flattening* and further analysis

Table 1: Supervised Classification Results on CIFAR-10 (a) and CIFAR-100 (b). We note significant improvement with *Manifold Mixup* especially in terms of Negative log-likelihood (NLL). Please refer to Appendix C for details on the implementation of *Manifold Mixup* and *Manifold Mixup* All layers and results on SVHN. † and ‡ refer to the results reported in (Zhang et al., 2018) and (Guo et al., 2016) respectively.

| Model | Test Error | Test NLL |
|---|---|---|
| PreActResNet18 | | |
| No Mixup | 5.12 | 0.2646 |
| Input Mixup ($\alpha = 1.0$) † | 3.90 | n/a |
| AdaMix ‡ | 3.52 | n/a |
| Input Mixup ($\alpha = 1.0$) | 3.50 | 0.1945 |
| *Manifold Mixup* ($\alpha = 2.0$) | **2.89** | **0.1407** |
| PreActResNet152 | | |
| No Mixup | 4.20 | 0.1994 |
| Input Mixup ($\alpha = 1.0$) | 3.15 | 0.2312 |
| *Manifold Mixup* ($\alpha = 2.0$) | 2.76 | 0.1419 |
| *Manifold Mixup* all layers ($\alpha = 6.0$) | **2.38** | **0.0957** |

(a) CIFAR-10

| Model | Test Error | Test NLL |
|---|---|---|
| PreActResNet18 | | |
| No Mixup † | 25.60 | n/a |
| No Mixup | 24.68 | 1.284 |
| Input Mixup ($\alpha = 1.0$) † | 21.10 | n/a |
| AdaMix ‡ | **20.97** | n/a |
| *Manifold Mixup* ($\alpha = 2.0$) | 21.05 | **0.913** |
| PreActResNet34 | | |
| Input Mixup ($\alpha = 1.0$) | 22.79 | 1.085 |
| *Manifold Mixup* ($\alpha = 2.0$) | **20.39** | **0.930** |

(b) CIFAR-100

*Manifold Mixup* is that both consider linear interpolations in hidden states with the same interpolation applied to the labels. However an important difference is that *Manifold Mixup* backpropagates gradients through the earlier parts of the network (the layers before where mixup is applied) unlike Zhao & Cho (2018). As discussed in Section 5.1 and Appendix B this was found to significantly change the learning process.

AdaMix (Guo et al., 2018a) is another related method which attempted to learn better mixing distributions to avoid overlap. AdaMix reported 3.52% error on CIFAR-10 and 20.97% error on CIFAR-100. We report 2.38% error on CIFAR-10 and 20.39% error on CIFAR-100. AdaMix only interpolated in the input space, and they report that their method hurt results significantly when they tried to apply it to the hidden layers. Thus this method likely works for different reasons from *Manifold Mixup* and might be complementary.

AgrLearn (Guo et al., 2018b) is a method which adds a new information bottleneck layer to the end of deep neural networks. This achieved substantial improvements, and was used together with Input Mixup (Zhang et al., 2018) to achieve 2.45% test error on CIFAR-10. As their method was complimentary with Input Mixup, it's possible that their method is also complimentary with *Manifold Mixup*, and this could be an interesting area for future work.

## 5 EXPERIMENTS

### 5.1 REGULARIZATION ON SUPERVISED LEARNING

We present results on *Manifold Mixup* based regularization of networks using the PreActResNet architecture (He et al., 2016). We closely followed the procedure of (Zhang et al., 2018) as a way of providing direct comparisons with the Input Mixup algorithm. We used weight decay of 0.0001 and trained with SGD with momentum and multiplied the learning rate by 0.1 at regularly scheduled epochs. These results for CIFAR-10 and CIFAR-100 are in Table 1a and 1b. We also ran experiments where we took PreActResNet34 models trained on the normal CIFAR-100 data and evaluated them on test sets with artificial deformations (shearing, rotation, and zooming) and showed that *Manifold Mixup* demonstrated significant improvements (Appendix C Table 5), which suggests that *Manifold Mixup* performs better on the variations in the input space not seen during the training. We also show that the number of epochs needed to reach good results is not significantly affected by using *Manifold Mixup* in Figure 8.

To better understand why the method works, we performed an experiment where we trained with *Manifold Mixup* but blocked gradients immediately after the layer where we perform mixup. On CIFAR-10 PreActResNet18, this caused us to achieve 4.86% test error when trained on 400 epochs and 4.33% test error when trained on 1200 epochs. This is better than the baseline, but worse than *Manifold Mixup* or Input Mixup in both cases. Because we randomly select the layer to mix, each layer of the network is still being trained, although not on every update. This demonstrates that the *Manifold Mixup* method improves results by changing the layers both before and after the mixup operation is applied.

We also compared *Manifold Mixup* against other strong regularizers. We selected the best performing hyperparameters for each of the following models using a validation set. Using each model's best performing hyperparameters, test error averages and standard deviations for five trials (in %) for CIFAR-10 using PreResNet50 trained for 600 epochs are: vanilla PreResNet50 ($4.96 \pm 0.19$), Dropout ($5.09 \pm 0.09$), Cutout (Devries & Taylor, 2017) ($4.77 \pm 0.38$), Mixup ($4.25 \pm 0.11$) and Manifold Mixup ($3.77 \pm 0.18$). This clearly shows that *Manifold Mixup* has strong regularizing effects. (Note that the results in Table 1 were run for 1200 epochs and thus these results are not directly comparable.)

We also evaluate the quality of the representations learned by *Manifold Mixup* by applying K-Nearest Neighbour classifier on the feature extracted from the top layer of PreResNet18 for CIFAR-10. We achieved test errors of 6.09% (Vanilla PreResNet18), 5.54% (Mixup) and 5.16% (Manifold Mixup). It suggests that *Manifold Mixup* helps learning better representations. Further analysis of how *Manifold Mixup* changes the representations is given in Appendix B

There are a couple of important questions to ask: how sensitive is the performance of *Manifold Mixup* with respect to the hyperparameter $\alpha$ and in which layers the mixing should be performed. We found that *Manifold Mixup* works well for a wide range of $\alpha$ values. Please refer to Appendix J for more details. Furthermore, the results in Appendix K suggests that mixing should not be performed in the layers very close to the output layer.

## 5.2 Semi-Supervised Learning

Semi-supervised learning is concerned with building models which can take advantage of both labeled and unlabeled data. It is particularly useful in domains where obtaining labels is challenging, but unlabeled data is plentiful.

The *Manifold Mixup* approach to semi-supervised learning is closely related to the consistency regularization approach reviewed by Oliver et al. (2018). It involves minimizing loss on labelled samples as well as unlabeled samples by controlling the trade-off between these two losses via a consistency coefficient. In the *Manifold Mixup* approach for semi-supervised learning, the loss from labeled examples is computed as normal. For computing loss from unlabelled samples, the model's predictions are evaluated on a random batch of unlabeled data points. Then the normal manifold mixup procedure is used, but the targets to be mixed are the soft target outputs from the classifier. The detailed algorithm for both *Manifold Mixup* and Input Mixup with semi-supervised learning are given in appendix D.

Table 2: Results on semi-supervised learning (SSL) on CIFAR-10 (4k labels) and SVHN (1k labels) (in test error %). All results use the same standardized architecture (WideResNet-28-2). Each experiment was run for 5 trials. † refers to the results reported in Oliver et al. (2018)

| SSL Approach | CIFAR-10 | SVHN |
|---|---|---|
| Supervised † | $20.26 \pm 0.38$ | $12.83 \pm 0.47$ |
| Mean-Teacher † | $15.87 \pm 0.28$ | $5.65 \pm 0.47$ |
| VAT † | $13.86 \pm 0.27$ | $5.63 \pm 0.20$ |
| VAT-EM † | $13.13 \pm 0.39$ | $\mathbf{5.35 \pm 0.19}$ |
| Semi-supervised Input Mixup | $10.71 \pm 0.44$ | $6.54 \pm 0.62$ |
| Semi-supervised *Manifold Mixup* | $\mathbf{10.26 \pm 0.32}$ | $5.70 \pm 0.48$ |

Oliver et al. (2018) performed a systematic study of semi-supervised algorithms using a fixed wide resnet architecture "WRN-28-2" (Zagoruyko & Komodakis, 2016). We evaluate *Manifold Mixup* using this same setup and achieve improvements for CIFAR-10 over the previously best performing

algorithm, Virtual Adversarial Training (VAT) (Miyato et al., 2018a) and Mean-Teachers (Tarvainen & Valpola, 2017). For SVHN, *Manifold Mixup* is competitive with VAT and Mean-Teachers. See Table 2. While VAT requires an additional calculation of the gradient and Mean-Teachers requires repeated model parameters averaging, *Manifold Mixup* requires no additional (non-trivial) computation.

In addition, we also explore the regularization ability of *Manifold Mixup* in a fully-supervised low-data regime by training a PreResnet-152 model on 4000 labeled images from CIFAR-10. We obtained 13.64 % test error which is comparable with the fully-supervised regularized baseline according to results reported in Oliver et al. (2018). Interestingly, we do not use a combination of two powerful regularizers ("Shake-Shake" and "Cut-out") and the more complex ResNext architecture as in Oliver et al. (2018) and still achieve the same level of test accuracy, while doing much better than the fully supervised baseline not regularized with state-of-the-art regularizers (20.26% error).

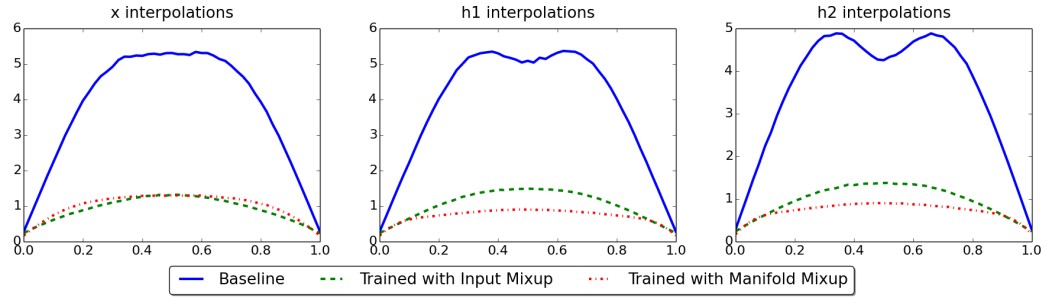

Figure 2: Study of test Negative Log-likelihood (NLL) using the interpolated target values (lower is better) on interpolated points under models trained with the baseline, mixup, and *Manifold Mixup*. *Manifold Mixup* dramatically improves performance when interpolating in the hidden states, and very slightly reduces performance when interpolating in the visible space. Y-axis denotes NLL and X-axis denotes the interpolation coefficient

## 5.3 ADVERSARIAL EXAMPLES

Adversarial examples in some sense are the "worst case" scenario for models failing to perform well when evaluated with data off the manifold[3]. Because *Manifold Mixup* only considers a subset of directions around data points (namely, those corresponding to interpolations), we would not expect the model to be robust to adversarial attacks which can consider any direction within an epsilon-ball of each example. At the same time, *Manifold Mixup* expands the set of points seen during training, so an intriguing hypothesis is that these overlap somewhat with the set of possible adversarial examples, which would force adversarial attacks to consider a wider set of directions, and potentially be more computationally expensive. To explore this we considered the Fast Gradient Sign Method (FGSM, Goodfellow et al., 2015) which only requires a single gradient update and considers a relatively small subset of adversarial directions. The resulting performance of *Manifold Mixup* against FGSM are given in Table 3. A challenge in evaluating adversarial examples comes from the gradient masking problem in which a defense succeeds solely due to reducing the quality of the gradient signal. Athalye et al. (2018) explored this issue in depth and proposed running an unbounded search for a large number of iterations to confirm the quality of the gradient signal. Our *Manifold Mixup* passed this sanity check (see Appendix F). While we found that *Manifold Mixup* greatly improved robustness to the FGSM attack, especially over Input Mixup (Zhang et al., 2018), we found that *Manifold Mixup* did not significantly improve robustness against the stronger iterative projected gradient descent (PGD) attack (Madry et al., 2018).

## 6 VISUALIZATION OF INTERPOLATED STATES

An important question is what kinds of feature combinations are being explored when we perform mixup in the hidden layers as opposed to linear interpolation in visible space. To provide a qualita-

---

[3]See the adversarial spheres (Gilmer et al., 2018) paper for a discussion of what it means to be off of the manifold.

Table 3: CIFAR-10 Test Accuracy Results on white-box FGSM (Goodfellow et al., 2015) adversarial attack (higher is better) using PreActResNet18 (left). SVHN Test Accuracy Results on white-box FGSM using WideResNet20-10 (Zagoruyko & Komodakis, 2016). Note that our method achieves some degree of adversarial robustness, against the FGSM attack, despite not requiring any additional (significant) computation. † refers to results reported in (Madry et al., 2018)

| CIFAR-10 Models | FGSM |
|---|---|
| Adv. Training (PGD 7-step) † Adversarial Training | 56.10 |
| + Fortified Networks | 81.80 |
| Baseline (Vanilla Training) | 36.32 |
| Input Mixup ($\alpha = 1.0$) | 71.51 |
| *Manifold Mixup* ($\alpha = 2.0$) | 77.50 |

| CIFAR-100 Models | FGSM |
|---|---|
| Input Mixup ($\alpha = 1.0$) | 40.7 |
| *Manifold Mixup* ($\alpha = 2.0$) | 44.96 |

| SVHN Models | FGSM |
|---|---|
| Baseline | 21.49 |
| Input Mixup ($\alpha = 1.0$) | 56.98 |
| *Manifold Mixup* ($\alpha = 2.0$) | 65.91 |
| Adv. Training (PGD 7-step) | 72.80 |

tive study of this, we trained a small decoder convnet (with upsampling layers) to predict an image from the *Manifold Mixup* classifier's hidden representation (using a simple squared error loss in the visible space). We then performed mixup on the hidden states between two random examples, and ran this interpolated hidden state through the convnet to get an estimate of what the point would look like in input space. Similarly to earlier results on auto-encoders (Bengio et al., 2013), we found that these interpolated $h$ points corresponded to images with a blend of the features from the two images, as opposed to the less-semantic pixel-wise blending resulting from Input Mixup as shown in Figure 3 and Figure 4. Furthermore, this justifies the training objective for examples mixed-up in the hidden layers: (1) most of the interpolated points correspond to combinations of semantically meaningful factors, thus leading to the more training samples; and (2) none of the interpolated points between objects of two different categories A and B correspond to a third category C, thus justifying a training target which gives 0 probability on all the classes except A and B.

## 7 CONCLUSION

Deep neural networks often give incorrect yet extremely confident predictions on data points which are unlike those seen during training. This problem is one of the most central challenges in deep learning both in theory and in practice. We have investigated this from the perspective of the representations learned by deep networks. In general, deep neural networks can learn representations such that real data points are widely distributed through the space and most of the area corresponds to high confidence classifications. This has major downsides in that it may be too easy for the network to provide high confidence classification on points which are off of the data manifold and also that it may not provide enough incentive for the network to learn highly discriminative representations. We have presented *Manifold Mixup*, a new algorithm which aims to improve the representations learned by deep networks by encouraging most of the hidden space to correspond to low confidence classifications while concentrating the hidden states for real examples onto a lower dimensional subspace. We applied *Manifold Mixup* to several tasks and demonstrated improved test accuracy and dramatically improved test likelihood on classification, better robustness to adversarial examples from FGSM attack, and improved semi-supervised learning. *Manifold Mixup* incurs virtually no additional computational cost, making it appealing for practitioners.

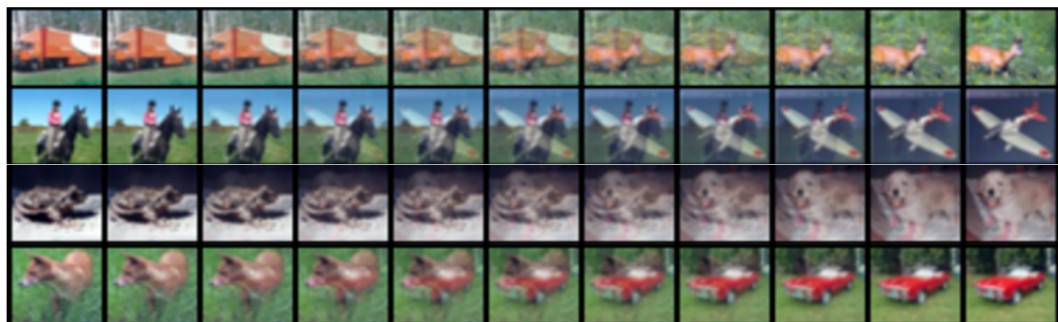

Figure 3: **Interpolations in the input space** with a mixing rate varied from 0.0 to 1.0.

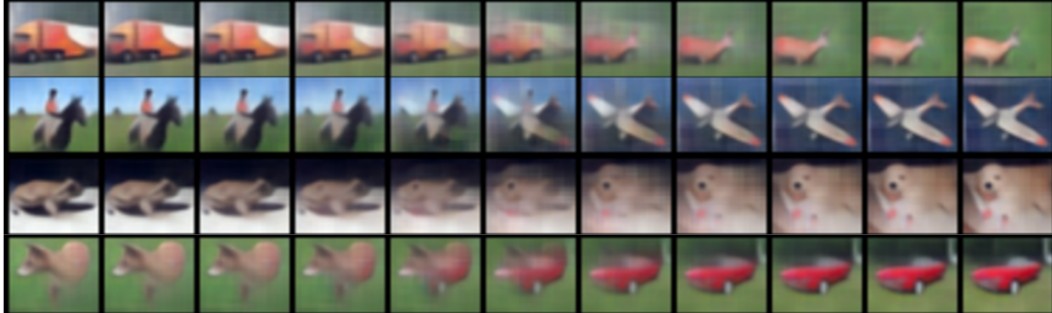

Figure 4: **Interpolations in the hidden states** (using a small convolutional network trained to predict the input from the output of the second resblock). The interpolations in the hidden states show a better blending of semantically relevant features, and more of the images are visually consistent.

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

## A  Synthetic Experiments Analysis

We conducted experiments using a generated synthetic dataset where each image is deterministically rendered from a set of independent factors. The goal of this experiment is to study the impact of input mixup and an idealized version of *Manifold Mixup* where we know the true factors of variation in the data and we can do mixup in exactly the space of those factors. This is not meant to be a fair evaluation or representation of how *Manifold Mixup* actually performs - rather it's meant to illustrate how generating relevant and semantically meaningful augmented data points can be much better than generating points which are far off the data manifold.

We considered three tasks. In Task A, we train on images with angles uniformly sampled between $(-70°, -50°)$ (label 0) with 50% probability and uniformly between $(50°, 80°)$ (label 1) with 50% probability. At test time we sampled uniformly between $(-30°, -10°)$ (label 0) with 50% probability and uniformly between $(10°, 30°)$ (label 1) with 50% probability. Task B used the same setup as Task A for training, but the test instead used $(-30°, -20°)$ as label 0 and $(-10°, 30°)$ as label 1. In Task C we made the label whether the digit was a "1" or a "7", and our training images were uniformly sampled between $(-70°, -50°)$ with 50% probability and uniformly between $(50°, 80°)$ with 50% probability. The test data for Task C were uniformly sampled with angles from $(-30°, 30°)$.

The examples of the data are in figure 5 and results are in table 4. In all cases we found that Input Mixup gave some improvements in likelihood but limited improvements in accuracy - suggesting that the even generating nonsensical points can help a classifier trained with Input Mixup to be better calibrated. Nonetheless the improvements were much smaller than those achieved with mixing in the ground truth attribute space.

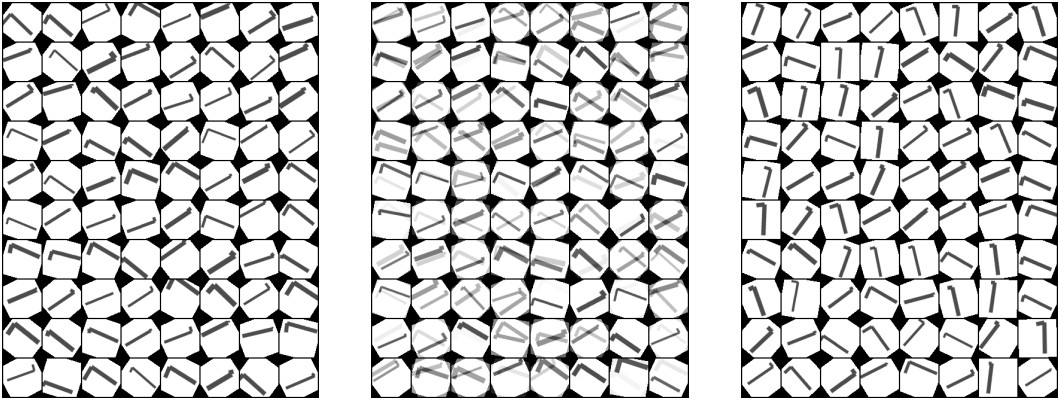

Figure 5: Synthetic task where the underlying factors are known exactly. Training images (left), images from input mixup (center), and images from mixing in the ground truth factor space (right).

Table 4: Results on synthetic data generalization task with an idealized Manifold Mixup (mixing in the true latent generative factors space). Note that in all cases visible mixup significantly improved likelihood, but not to the same degree as factor mixup.

| Task | Model | Test Accuracy | Test NLL |
|---|---|---|---|
| **Task A** | No Mixup | 1.6 | 8.8310 |
| | Input Mixup (1.0) | 0.0 | 6.0601 |
| | Ground Truth Factor Mixup (1.0) | 94.77 | 0.4940 |
| **Task B** | No Mixup | 21.25 | 7.0026 |
| | Input Mixup (1.0) | 18.40 | 4.3149 |
| | Ground Truth Factor Mixup (1.0) | 84.02 | 0.4572 |
| **Task C** | No Mixup | 63.05 | 4.2871 |
| | Input Mixup | 66.09 | 1.4181 |
| | Ground Truth Factor Mixup | 99.06 | 0.1279 |

# B    ANALYSIS OF HOW *Manifold Mixup* CHANGES LEARNED REPRESENTATIONS

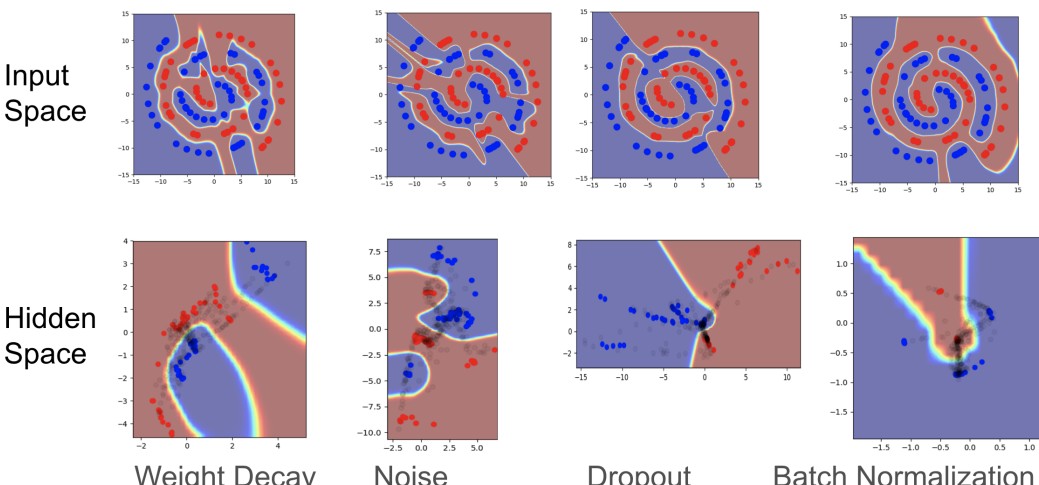

Figure 6: An experiment on a network trained on the 2D spiral dataset with a 2D bottleneck hidden state in the middle of the network (the same setup as 1). Noise refers to gaussian noise in the bottleneck layer, dropout refers to dropout of 50% in all layers except the bottleneck itself (due to its low dimensionality), and batch normalization refers to batch normalization in all layers. This shows that the effect of concentrating the hidden states for each class and providing a broad region of low confidence between the regions is not accomplished by the other regularizers.

We have found significant improvements from using *Manifold Mixup*, but a key question is whether the improvements come from changing the behavior of the layers before the mixup operation is applied or the layers after the mixup operation is applied. This is a place where *Manifold Mixup* and Input Mixup are clearly differentiated, as Input Mixup has no "layers before the mixup operation" to change. We conducted analytical experimented where the representations are low-dimensional enough to visualize. More concretely, we trained a fully connected network on MNIST with two fully-connected leaky relu layers of 1024 units, followed by a 2-dimensional bottleneck layer, followed by two more fully-connected leaky-relu layers with 1024 units.

We then considered training with no mixup, training with mixup in the input space, and training *only* with mixup directly following the 2D bottleneck. We consistently found that *Manifold Mixup* has the effect of making the representations much tighter, with the real data occupying more specific points, and with a more well separated margin between the classes, as shown in Figure 7

# C    SUPERVISED REGULARIZATION

For supervised regularization we considered architectures within the PreActResNet family: PreActResNet18, PreActResNet34, and PreActResNet152. When using *Manifold Mixup*, we selected the layer to perform mixing uniformly at random from a set of eligible layers. In our experiments on PreActResNets in Table 1a, Table 1b, Table 6, Table 3 and Table 5, for *Manifold Mixup*, our eligible layers for mixing were : the input layer, the output from the first resblock, and the output from the second resblock. For PreActResNet18, the first resblock has four layers and the second resblock has four layers. For PreActResNet34, the first resblock has six layers and the second resblock has eight layers. For PreActResNet152, the first resblock has 9 layers and the second resblock has 24 layers. Thus the mixing is often done fairly deep in the network, for example in PreActResNet152 the output of the second resblock is preceded by a total of 34 layers (including the initial convolution which is not in a resblock). For *Manifold Mixup* All layers in Table 1a, our eligible layers for mixing were : the input layer, the output from the first resblock, and the output from the second resblock, and the output from the third resblock. We trained all models for 1200 epochs and dropped the learning rates by a factor of 0.1 at 400 epochs and 800 epochs.

Table 6 presents results for SVHN dataset with PreActResNet18 architecture.

In Figure 9 and Figure 10, we present the training loss (Binary cross entropy) for Cifar10 and Cifar100 datasets respectively. We observe that performing *Manifold Mixup* in higher layers allows the train loss to go down faster as compared against the Input Mixup. This is consistent with the demonstration in Figure 1: Input mixup

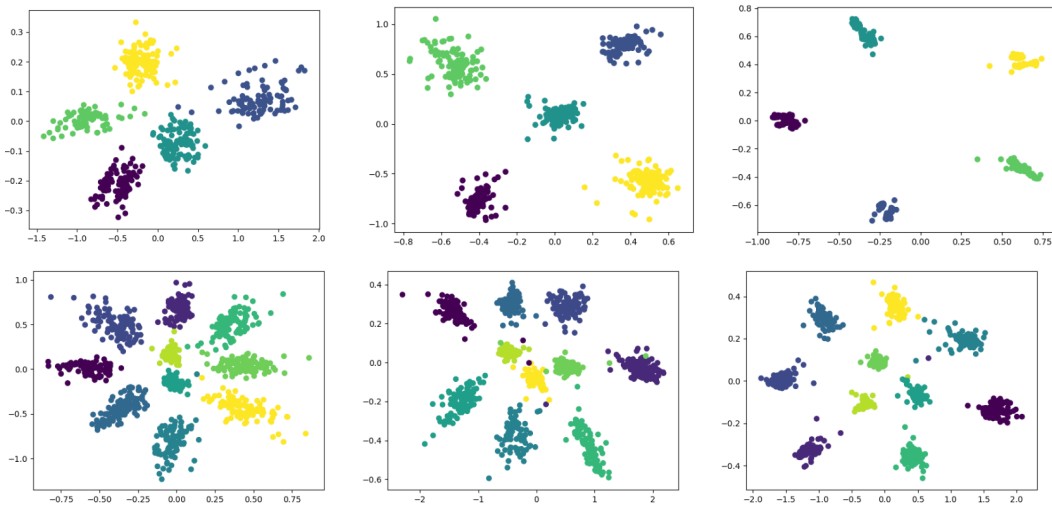

Figure 7: Representations from a classifier on MNIST (top is trained on digits 0-4, bottom is trained on all digits) with a 2D bottleneck representation in the middle layer. No Mixup Baseline (left), Input Mixup (center), *Manifold Mixup* (right).

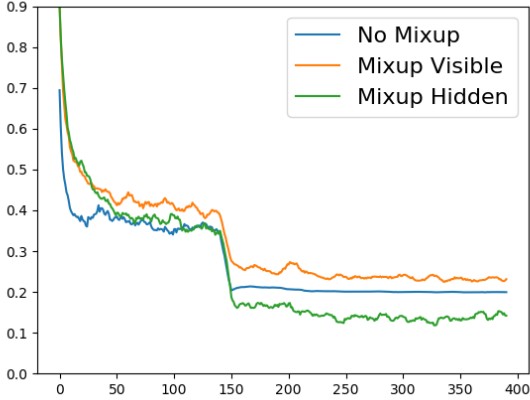

Figure 8: CIFAR-10 test set Negative Log-Likelihood (Y-axis) on PreActResNet152, wrt training epochs (X-axis).

can suffer from underfitting since the interpolation between two examples can intersect with a real example. In *Manifold Mixup* the hidden states in which the interpolation is performed, are learned, hence during the course of training they can evolve in such a way that the aforementioned intersection issue is avoided.

## D    SEMI-SUPERVISED MANIFOLD MIXUP AND INPUT MIXUP ALGORITHM

We present the procedure for Semi-supervised Manifold Mixup and Semi-supervised Input Mixup in Algorithms 1 and 3 respectively.

**Algorithm 1** Semi-supervised Manifold Mixup. $f_\theta$: Neural Network; $ManifoldMixup$: Manifold Mixup Algorithm 2; $D_L$: set of labelled samples; $D_{UL}$: set of unlabelled samples; $\pi$: consistency coefficient (weight of unlabeled loss, which is ramped up to increase from zero to its max value over the course of training); $N$: number of updates; $\tilde{y}_i$: Mixedup labels of labelled samples; $\hat{y}_i$: predicted label of the labelled samples mixed at a hidden layer; $y_j$: Psuedolabels for unlabelled samples; $\tilde{y}_j$: Mixedup Psuedolabels of unlabelled samples; $\hat{y}_j$ predicted label of the unlabelled samples mixed at a hidden layer

---

1:  $k \leftarrow 0$
2:  **while** $k \leq N$ **do**
3:      Sample $(x_i, y_i) \sim D_L$                                     ▷ Sample labeled batch
4:      $\hat{y}_i, \tilde{y}_i = ManifoldMixup(x_i, y_i, \theta)$
5:      $L_S = Loss(\hat{y}_i, \tilde{y}_i)$                              ▷ Cross Entropy loss
6:      Sample $x_j \sim D_{UL}$                                         ▷ Sample unlabeled batch
7:      $y_j = f_\theta(x_j)$                                            ▷ Compute Pseudolabels
8:      $\hat{y}_j, \tilde{y}_j = ManifoldMixup(x_j, y_j, \theta)$
9:      $L_{US} = Loss(\hat{y}_j, \tilde{y}_j)$                          ▷ MSE Loss
10:     L $= L_S + \pi(k)L_{US}$                                         ▷ Total Loss
11:     $g \leftarrow \nabla_\theta L$ (Gradients of the minibatch Loss )
12:     $\theta \leftarrow$ Update parameters using gradients $g$ (e.g. SGD )
13: **end while**

---

**Algorithm 2** Manifold Mixup. $f_\theta$: Neural Network; $D$ : dataset

---

1:  Sample $(x_i, y_i) \sim D$                                          ▷ Sample a batch
2:  $h_i \leftarrow$ hidden state representation of Neural Network $f_\theta$ at a layer $k$      ▷ the layer $k$ is chosen randomly
3:  $(h_i^{mixed}, y_i^{mixed}) \leftarrow Mixup(h_i, y_i)$
4:  $\hat{y}_i \leftarrow$ Forward Pass the $h_i^{mixed}$ from layer $k$ to the output layer of $f_\theta$
5:  return $\hat{y}_i, \tilde{y}_i$

---

**Algorithm 3** Semi-supervised Input Mixup. $f_\theta$: Neural Network. $InputMixup$: Mixup process of (Zhang et al., 2018); $D_L$: set of labelled samples; $D_{UL}$: set of unlabelled samples; $\pi$: consistency coefficient (weight of unlabeled loss, which is ramped up to increase from zero to its max value over the course of training); $N$: number of updates; $x_i^{mixedup}$: mixed up sample; $y_i^{mixedup}$: mixed up label; $\hat{y}_i^{mixedup}$: mixed up predicted label

---

1:  $k \leftarrow 0$
2:  **while** $k \leq N$ **do**
3:      Sample $(x_i, y_i) \sim D_L$
4:                                                                       ▷ Sample labeled batch
5:      $(x_i^{mixedup}, y_i^{mixedup}) = InputMixup(x_i, y_i)$
6:      $L_S = Loss(f_\theta(x_i^{mixedup}), y_i^{mixedup})$             ▷ CrossEntropy Loss
7:      Sample $x_j \sim D_{UL}$                                        ▷ Sample unlabeled batch
8:      $\hat{y}_j = f_\theta(x_j)$                                     ▷ Compute Pseudolabels
9:      $(x_j^{mixedup}, \hat{y}_j^{mixedup}) = InputMixup(x_j, \hat{y}_j)$
10:     $L_{US} = Loss(f_\theta(x_j^{mixedup}), \hat{y}_i^{mixedup})$   ▷ MSE Loss
11:     L $= L_S + \pi(k) * L_{US}$                                     ▷ Total Loss
12:     $g \leftarrow \nabla_\theta L$                                  ▷ Gradients of the minibatch Loss
13:     $\theta \leftarrow$ Update parameters using gradients $g$ (e.g. SGD )
14: **end while**

---

Table 5: Models trained on the normal CIFAR-100 and evaluated on a test set with novel deformations. *Manifold Mixup* (ours) consistently allows the model to be more robust to random shearing, rescaling, and rotation even though these deformations were not observed during training. For the rotation experiment, each image is rotated with an angle uniformly sampled from the given range. Likewise the shearing is performed with uniformly sampled angles. Zooming-in refers to take a bounding box at the center of the image with k% of the length and k% of the width of the original image, and then expanding this image to fit the original size. Likewise zooming-out refers to drawing a bounding box with k% of the height and k% of the width, and then taking this larger area and scaling it down to the original size of the image (the padding outside of the image is black).

| Test Set Deformation | No Mixup Baseline | Input Mixup $\alpha$=1.0 | Input Mixup $\alpha$=2.0 | *Manifold Mixup* $\alpha$=2.0 |
|---|---|---|---|---|
| Rotation U($-20°$,$20°$) | 52.96 | 55.55 | 56.48 | **60.08** |
| Rotation U($-40°$,$40°$) | 33.82 | 37.73 | 36.78 | **42.13** |
| Rotation U($-60°$,$60°$) | 26.77 | 28.47 | 27.53 | **33.78** |
| Rotation U($-80°$,$80°$) | 24.19 | 26.72 | 25.34 | **29.95** |
| Shearing U($-28.6°$, $28.6°$) | 55.92 | 58.16 | 60.01 | **62.85** |
| Shearing U($-57.3°$, $57.3°$) | 35.66 | 39.34 | 39.7 | **44.27** |
| Shearing U($-114.6°$, $114.6°$) | 19.57 | 22.94 | 22.8 | **24.69** |
| Shearing U($-143.2°$, $143.2°$) | 17.55 | 21.66 | 21.22 | **23.56** |
| Shearing U($-171.9°$, $171.9°$) | 22.38 | 25.53 | 25.27 | **28.02** |
| Zoom In (20% rescale) | 2.43 | 1.9 | **2.45** | 2.03 |
| Zoom In (40% rescale) | 4.97 | 4.47 | **5.23** | 4.17 |
| Zoom In (60% rescale) | 12.68 | **13.75** | 13.12 | 11.49 |
| Zoom In (80% rescale) | 47.95 | 52.18 | 50.47 | **52.7** |
| Zoom Out (120% rescale) | 43.18 | 60.02 | 61.62 | **63.59** |
| Zoom Out (140% rescale) | 19.34 | 41.81 | 42.02 | **45.29** |
| Zoom Out (160% rescale) | 11.12 | 25.48 | 25.85 | **27.02** |
| Zoom Out (180% rescale) | 7.98 | **18.11** | 18.02 | 15.68 |

Table 6: Results on SVHN dataset with PreActResNet18 architecture

| Model | Test Error ( in %) |
|---|---|
| PreActResNet18 | |
| No Mixup | 2.22 |
| Input Mixup ($\alpha = 0.01$) | 2.30 |
| Input Mixup ($\alpha = 0.05$) | 2.28 |
| Input Mixup ($\alpha = 0.2$) | 2.29 |
| Input Mixup ($\alpha = 0.5$) | 2.26 |
| Input Mixup ($\alpha = 1.0$) | 2.37 |
| Input Mixup ($\alpha = 1.5$) | 2.41 |
| *Manifold Mixup* ($\alpha = 1.5$) | 1.92 |
| *Manifold Mixup* ($\alpha = 2.0$) | **1.90** |

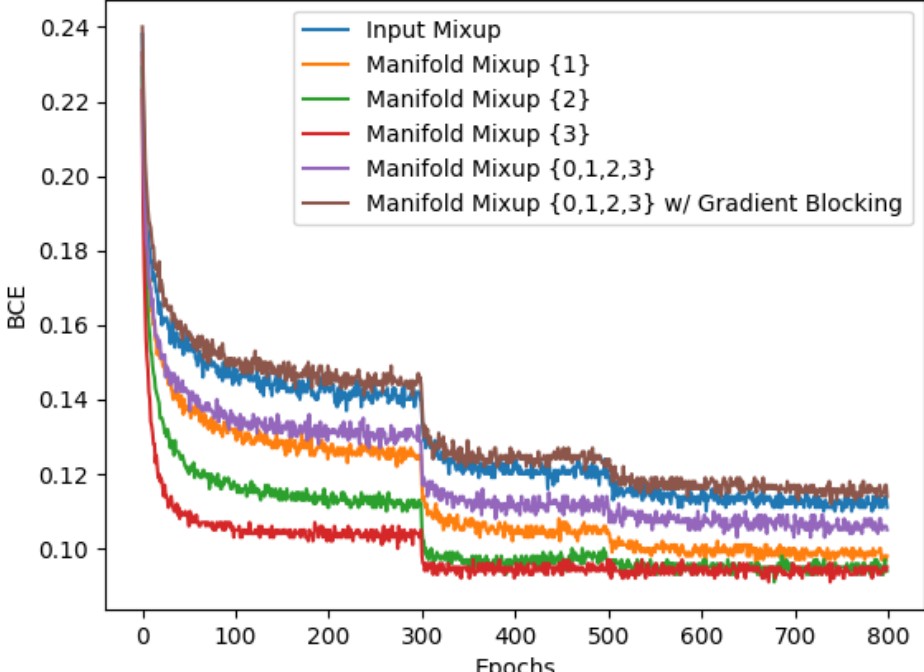

Figure 9: CIFAR-10 train set Binary Cross Entropy Loss (BCE) on Y-axis using PreActResNet18, with respect to training epochs (X-axis). The numbers in {} refer to the resblock after which *Manifold Mixup* is performed. The ordering of the losses is consistent over the course of training: Manifold Mixup with gradient blocked before the mixing layer has the highest training loss, followed by Input Mixup. The lowest training loss is achieved by mixing in the deepest layer, which is highly consistent with Section 3 which suggests that having more hidden units can help to prevent under-fitting.

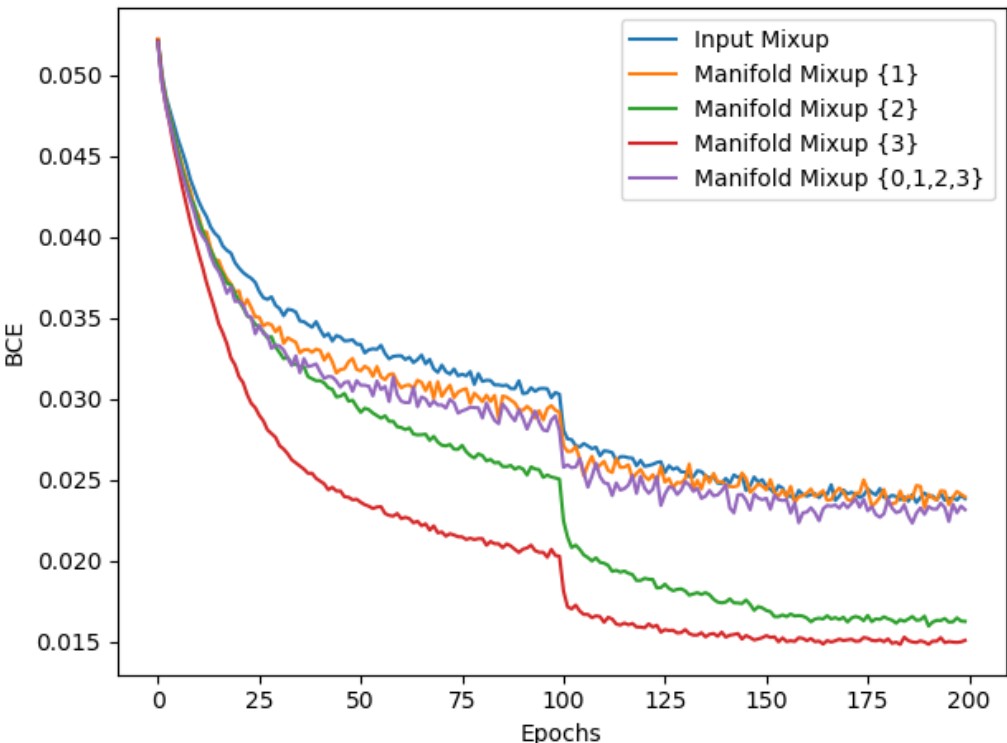

Figure 10: CIFAR-100 train set Binary Cross Entropy Loss (BCE) on Y-axis using PreActRes-Net50, with respect to training epochs (X-axis). The numbers in {} refer to the resblock after which *Manifold Mixup* is performed. The lowest training loss is achieved by mixing in the deepest layer.

## E    SEMI-SUPERVISED EXPERIMENTAL DETAILS

We use the WideResNet28-2 architecture used in (Oliver et al., 2018) and closely follow their experimental setup for fair comparison with other Semi-supervised learning algorithms. We used SGD with momentum optimizer in our experiments. For Cifar10, we run the experiments for 1000 epochs with initial learning rate is 0.1 and it is annealed by a factor of 0.1 at epoch 500, 750 and 875. For SVHN, we run the experiments for 200 epochs with initial learning rate is 0.1 and it is annealed by a factor of 0.1 at epoch 100, 150 and 175. The momentum parameter was set to 0.9. We used L2 regularization coefficient 0.0005 and L1 regularization coefficient 0.001 in our experiments. We use the batch-size of 100.

The data pre-processing and augmentation in exactly the same as in (Oliver et al., 2018). For CIFAR-10, we use the standard train/validation split of 45,000 and 5000 images for training and validation respectively. We use 4000 images out of 45,000 train images as labelled images for semi-supervised learning. For SVHN, we use the standard train/validation split with 65932 and 7325 images for training and validation respectively. We use 1000 images out of 65932 images as labelled images for semi-supervised learning. We report the test accuracy of the model selected based on best validation accuracy.

For supervised loss, we used $\alpha$ (of $\lambda \sim Beta(\alpha, \alpha)$) from the set { 0.1, 0.2, 0.3... 1.0} and found 0.1 to be the best. For unsupervised loss, we used $\alpha$ from the set {0.1, 0.5, 1.0, 1.5, 2.0. 3.0, 4.0} and found 2.0 to be the best.

The consistency coefficient is ramped up from its initial value 0.0 to its maximum value at 0.4 factor of total number of iterations using the same sigmoid schedule of (Tarvainen & Valpola, 2017). For CIFAR-10, we found max consistency coefficient = 1.0 to be the best. For SVHN, we found max consistency coefficient = 2.0 to be the best.

When using *Manifold Mixup*, we selected the layer to perform mixing uniformly at random from a set of eligible layers. In our experiments on WideResNet28-2 in Table 2, our eligible layers for mixing were : the input layer, the output from the first resblock, and the output from the second resblock.

## F    ADVERSARIAL EXAMPLES

We ran the unbounded projected gradient descent (PGD) (Madry et al., 2018) sanity check suggested in (Athalye et al., 2018). We took our trained models for the input mixup baseline and manifold mixup and we ran PGD for 200 iterations with a step size of 0.01 which reduced the mixup model's accuracy to 1% and reduced the *Manifold Mixup* model's accuracy to 0%. This is direct evidence that our defense did not improve results primarily as a result of gradient masking.

The Fast Gradient Sign Method (FGSM)  Goodfellow et al. (2015) is a simple one-step attack that produces $\widetilde{x} = x + \varepsilon \operatorname{sgn}(\nabla_x L(\theta, x, y))$.

## G    GENERATIVE ADVERSARIAL NETWORKS

The recent literature has suggested that regularizing the discriminator is beneficial for training GANs (Salimans et al., 2016; Arjovsky et al., 2017; Gulrajani et al., 2017; Miyato et al., 2018b). In a similar vein, one could add mixup to the original GAN training objective such that the extra data augmentation acts as a beneficial regularization to the discriminator, which is what was proposed in Zhang et al. (2018). Mixup proposes the following objective[4]:

$$\max_g \min_d \mathbb{E}_{x, z, \lambda} \, \ell(d(\lambda x_1 + (1 - \lambda)x_2), y(\lambda; x_1, x_2)), \qquad (8)$$

where $x_1, x_2$ can be either real or fake samples, and $\lambda$ is sampled from a $Uniform(0, \alpha)$. Note that we have used a function $y(\lambda; x_1, x_2)$ to denote the label since there are four possibilities depending on $x_1$ and $x_2$:

$$y(\lambda; x_1, x_2) = \begin{cases} \lambda, & \text{if } x_1 \text{ is real and } x_2 \text{ is fake} \\ 1 - \lambda, & \text{if } x_1 \text{ is fake and } x_2 \text{ is real} \\ 0, & \text{if both are fake} \\ 1, & \text{if both are real} \end{cases} \qquad (9)$$

In practice however, we find that it did not make sense to create mixes between real and real where the label is set to 1, (as shown in equation 9), since the mixup of two real examples in input space is not a real example.

---

[4]The formulation written is based on the official code provided with the paper, rather than the description in the paper. The discrepancy between the two is that the formulation in the paper only considers mixes between real and fake.

So we only create mixes that are either real-fake, fake-real, or fake-fake. Secondly, instead of using just the equation in 8, we optimize it in addition to the regular minimax GAN equations:

$$\max_g \min_d \mathbb{E}_x \, \ell(d(x), 1) + \mathbb{E}_{g(z)} \, \ell(d(g(z)), 0) + \text{GAN mixup term (Equation 8)} \tag{10}$$

Using similar notation to earlier in the paper, we present the manifold mixup version of our GAN objective in which we mix in the hidden space of the discriminator:

$$\min_d \mathbb{E}_{x,z,k} \, \ell(d(x), 1) + \ell(d(g(z), 0) + \ell(d_k(\lambda h_k(x_1) + (1 - \lambda)h_k(x_2), y(\lambda; x_1, x_2)), \tag{11}$$

where $h_k(\cdot)$ is a function denoting the intermediate output of the discriminator at layer $k$, and $d_k(\cdot)$ the output of the discriminator given input from layer $k$.

The layer $k$ we choose the sample can be arbitrary combinations of the input layer (i.e., input mixup), or the first or second resblocks of the discriminator, all with equal probability of selection.

We run some experiments evaluating the quality of generated images on CIFAR10, using as a baseline JSGAN with spectral normalization (Miyato et al., 2018b) (our configuration is almost identical to theirs). Results are averaged over at least three runs[5]. From these results, the best-performing mixup experiments (both input and *Manifold Mixup*) is with $\alpha = 0.5$, with mixing in all layers (both resblocks and input) achieving an average Inception / FID of $8.04 \pm 0.08$ / $21.2 \pm 0.47$, input mixup achieving $8.03 \pm 0.08$ / $21.4 \pm 0.56$, for the baseline experiment $7.97 \pm 0.07$ / $21.9 \pm 0.62$. This suggests that mixup acts as a useful regularization on the discriminator, which is even further improved by *Manifold Mixup*. See Figure 11 for the full set of experimental results

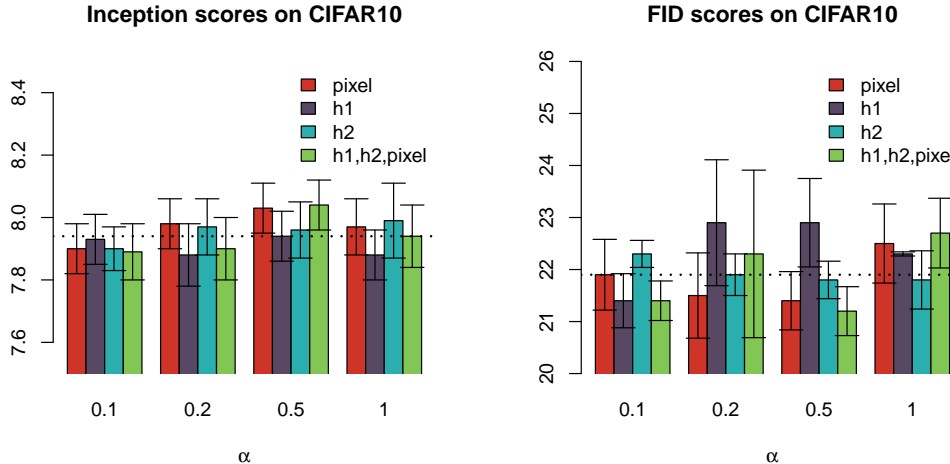

Figure 11: We test out various values of $\alpha$ in conjunction with either: input mixup (`pixel`) (Zhang et al., 2018), mixing in the output of the first resblock (`h1`), mixing in either the output of the first resblock or the output of the second resblock (`h1,2`), and mixing in the input or the output of the first resblock or the output of the second resblock (`1,2,pixel`). The dotted line indicates the baseline Inception / FID score. Higher scores are better for Inception, while lower is better for FID.

## H    INTUITIVE EXPLANATION OF HOW MANIFOLD MIXUP AVOIDS INCONSISTENT INTERPOLATIONS

An essential motivation behind manifold mixup is that because the network *learns* the hidden states, it can do so in such a way that the interpolations between points are consistent. Section 3 characterized this for hidden states with any number of dimensions and Figure 1 showed how this can occur on the 2d spiral dataset.

Our goal here is to discuss concrete examples to illustrate what it means for the interpolations to be consistent. If we consider any two points, the interpolated point between them is based on a sampled $\lambda$ and the soft-target

---

[5]Inception scores are typically reported with a mean and variance, though this is across multiple splits of samples across a single model. Since we run multiple experiments, we average their respective means and variances.

for that interpolated point is the targets interpolated with the same $\lambda$. So if we consider two points A,B which have the same label, it is apparent that every point on the line between A and B should have that same label with 100% confidence. If we consider two points A,B with different labels, then the point which is halfway between them will be given the soft-label of 50% the label of A and 50% the label of B (and so on for other $\lambda$ values).

It is clear that for many arrangements of data points, it is possible for a point in the space to be reached through distinct interpolations between different pairs of examples, and reached with different $\lambda$ values. Because the learned model tries to capture the distribution $p(y|h)$, it can only assign a single distribution over the label values to a single particular point (for example it could say that a point is 100% label A, or it could say that a point is 50% label A and 50% label B). Intuitively, these inconsistent soft-labels at interpolated points can be avoided if the states for each class are more concentrated and classes vary along distinct dimensions in the hidden space. The theory in Section 3 characterizes exactly what this concentration needs to be: that the representations for each class need to lie on a subspace of dimension equal to "number of hidden dimensions" - "number of classes" + 1.

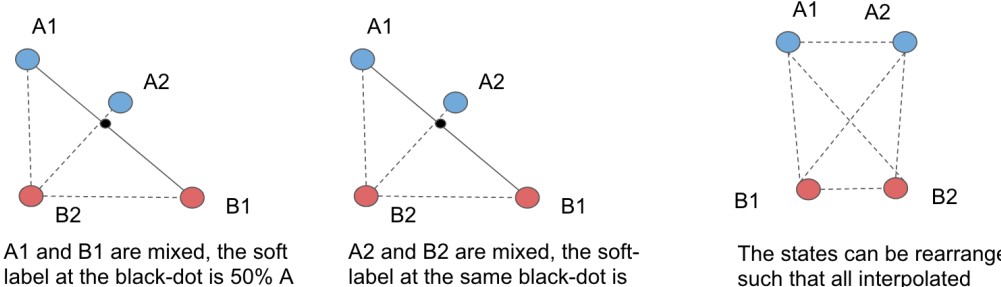

A1 and B1 are mixed, the soft label at the black-dot is 50% A and 50% B.

A2 and B2 are mixed, the soft-label at the same black-dot is ~95% A.

The states can be rearranged, such that all interpolated points give the same soft-label regardless of which two points were interpolated.

Figure 12: We consider a binary classification task with four data points represented in a 2D hidden space. If we perform mixup in that hidden space, we can see that if the points are laid out in a certain way, two different interpolations can give inconsistent soft-labels (left and middle). This leads to underfitting and high loss. When training with manifold mixup, this can be explicitly avoided because the states are *learned*, so the model can learn to produce states for which all interpolations give consistent labels, an example of which is seen on the right side of the figure.

# I  SPECTRAL ANALYSIS OF LEARNED REPRESENTATIONS

When we refer to *flattening*, we mean that the class-specific representations have reduced variability in some directions. Our analysis in this section makes this more concrete.

We trained an MNIST classifier with a hidden state bottleneck in the middle with 12 units (intentionally selected to be just slightly greater than the number of classes). We then took the representation for each class and computed a singular value decomposition (Figure 13 and Figure 14) and we also computed an SVD over all of the representations together (Figure 16). Our architecture contained three hidden layers with 1024 units and LeakyReLU activation, followed by a bottleneck representation layer (with either 12 or 30 hidden units), followed by an additional four hidden layers each with 1024 units and LeakyReLU activation. When we performed *Manifold Mixup* for our analysis, we only performed mixing in the bottleneck layer, and used a beta distribution with an alpha of 2.0. Additionally we performed another experiment (Figure 15 where we placed the bottleneck representation layer with 30 units immediately following the first hidden layer with 1024 units and LeakyReLU activation.

We found that *Manifold Mixup* had a striking effect on the singular values, with most of the singular values becoming much smaller. Effectively, this means that the representations for each class have variance in fewer directions. While our theory in Section 3 showed that this flattening must force each classes representations onto a lower-dimensional subspace (and hence an upper bound on the number of singular values) but this explores how this occurs empirically and does not require the number of hidden dimensions to be so small that it can be manually visualized. In our experiments we tried using 12 hidden units in the bottleneck Figure 13 as well as 30 hidden units Figure 14 in the bottleneck.

Our results from this experiment are unequivocal: *Manifold Mixup* dramatically reduces the size of the smaller singular values for each classes representations. This indicates a flattening of the class-specific representations. At the same time, the singular values over all the representations are not changed in a clear way (Figure 16), which suggests that this flattening occurs in directions which are distinct from the directions occupied by representations from other classes, which is the same intuition behind our theory. Moreover, Figure 15 shows that when the mixing is performed earlier in the network, there is still a flattening effect, though it is weaker than in the later layers, and again Input Mixup has an inconsistent effect.

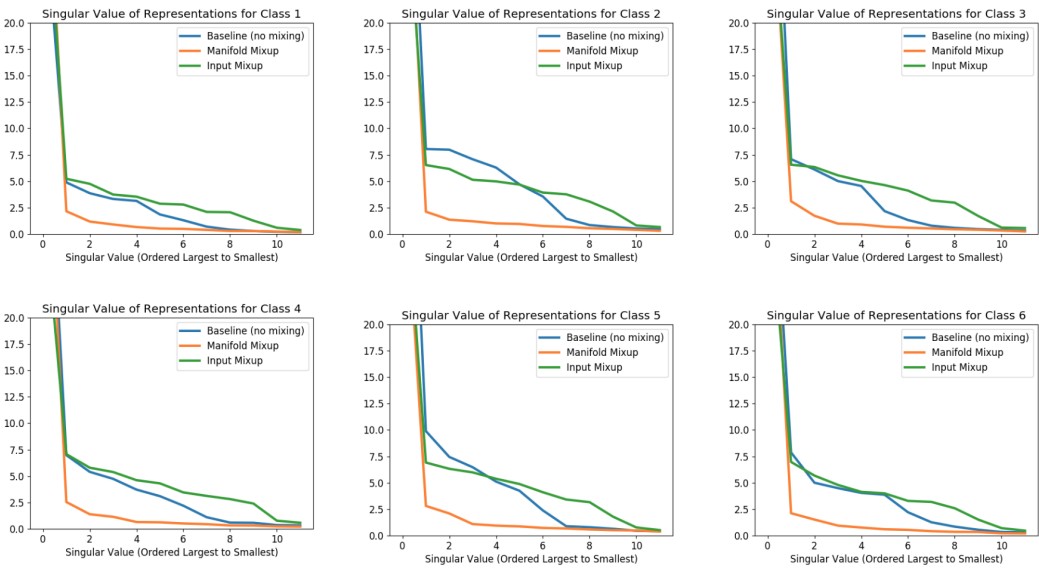

Figure 13: SVD on the class-specific representations in a bottleneck layer with 12 units following 3 hidden layers. For the first singular value, the value (averaged across the plots) is 50.08 for the baseline, 37.17 for Input Mixup, and 43.44 for *Manifold Mixup* (these are the values at x=0 which are cutoff). We can see that the class-specific SVD leads to singular values which are dramatically more concentrated when using *Manifold Mixup* with Input Mixup not having a consistent effect.

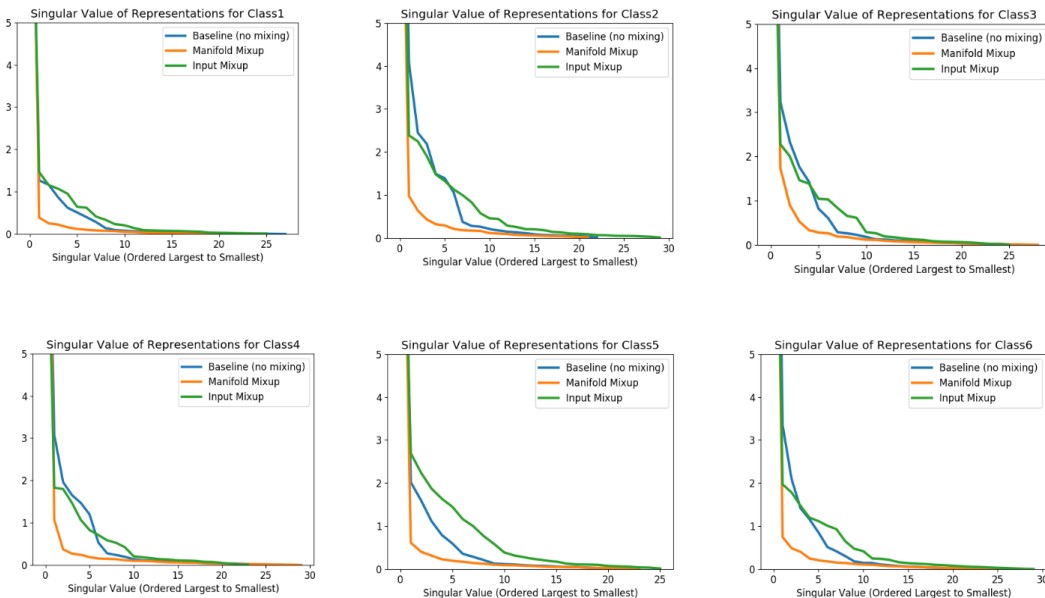

Figure 14: SVD on the class-specific representations in a bottleneck layer with 30 units following 3 hidden layers. For the first singular value, the value (averaged across the plots) is 14.68 for the baseline, 12.49 for Input Mixup, and 14.43 for *Manifold Mixup* (these are the values at x=0 which are cutoff).

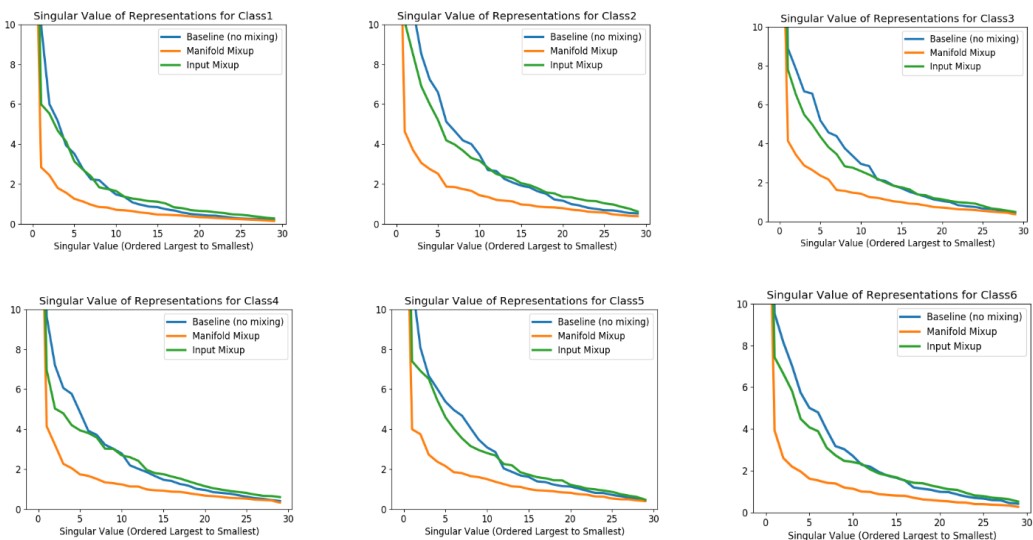

Figure 15: SVD on the class-specific representations in a bottleneck layer with 30 units following a single hidden layer. For the first singular value, the value (averaged across the plots) is 33.64 for the baseline, 27.60 for Input Mixup, and 24.60 for *Manifold Mixup* (these are the values at x=0 which are cutoff). We see that with the bottleneck layer placed earlier, the reduction in the singular values from *Manifold Mixup* is smaller but still clearly visible. This makes sense, as it is not possible for this early layer to be perfectly discriminative.

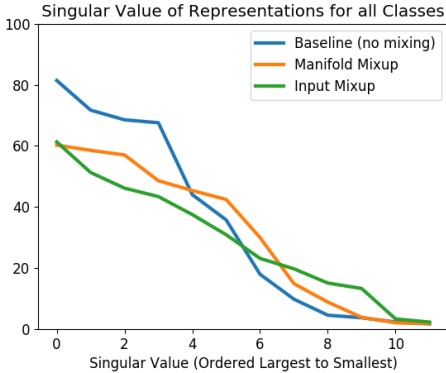

Figure 16: When we run SVD on all of the classes together (in the setup with 12 units in the bottleneck layer following 3 hidden layers), we see no clear difference in the singular values for the Baseline, Input Mixup, and *Manifold Mixup* models (ran on the model with a bottleneck hidden state of 12 dimensions). Thus we can see that the flattening effect of manifold mixup is entirely class-specific, and does not appear overall, which is consistent with what our theory has predicted.

## J    SENSITIVITY TO HYPER-PARAMETER $\alpha$

We compare the performance of *Manifold Mixup* using different values of hyper-parameter $\alpha$ by training a PreActResNet18 network on Cifar10 dataset, as shown in Table 7.

Table 7: Test accuracy (in %) of *Manifold Mixup* for a range of values for hyperparameter $\alpha$ for Cifar10 dataset on PreActReseNet18

| $\alpha$ | Input Mixup | Manifold Mixup |
|---|---|---|
| 0.5 | 95.75 | 96.12 |
| 1.0 | 95.84 | 96.10 |
| 1.2 | **96.09** | 96.29 |
| 1.5 | 96.06 | 96.35 |
| 1.8 | 95.97 | 96.45 |
| 2.0 | 95.83 | **96.73** |

Manifold Mixup outperformed Input Mixup for all alphas in the set (0.5, 1.0, 1.2, 1.5, 1.8, 2.0) - indeed the lowest result for Manifold Mixup is better than the worst result with Input Mixup. Note that Input Mixup's results deteriorate when using an alpha that is too large, which is not seen with manifold mixup.

## K    ABLATION STUDY FOR WHICH LAYER TO DO THE MIXING ON

In this section, we discuss which layers are a good candidate for mixing in the *Manifold Mixup* algorithm. We evaluated PreActResNet18 models on CIFAR-10 and considered mixing in a subset of the layers, we ran for fewer epochs than in the Section 5.1 (making the accuracies slightly lower across the board), and we decided to fix the alpha to 2.0 as we did in the the Section 5.1. We considered different subsets of layers to mix in, with 0 referring to the input layer, 1/2/3 referring to the output of the 1st/2nd/3rd resblocks respectively. For example 0,2 refers to mixing in the input layer and the output of the 2nd resblock. {} refers to no mixing. The results are presented in Table 8

Essentially, it helps to mix in more layers, except for the later layers which hurts the test accuracy to some extent - which is consistent with our theory in Section 3 : the theory in Section 3 assumes that the part of the network after mixing is a universal approximator, hence, there is a sensible case to be made for not mixing in the very last layers.

Table 8: Test accuracy (in %) of *Manifold Mixup* when the mixing is performed in different subsets of layers, for PreActReseNet18 on Cifar10 dataset.

| Layers | Test Accuracy |
| --- | --- |
| 0,1,2 | **96.73** |
| 0,1 | 96.40 |
| 0,1,2,3 | 96.23 |
| 1,2 | 96.14 |
| 0 | 95.83 |
| 1,2,3 | 95.66 |
| 1 | 95.59 |
| 2,3 | 94.63 |
| 2 | 94.31 |
| 3 | 93.96 |
| {} | 93.21 |

