# OpenReview forum: "Manifold Mixup: Learning Better Representations by Interpolating Hidden States"
_ICLR.cc/2019/Conference_

### Official Review · AnonReviewer1 · 2018-10-29
**Good paper**

**Rating:** 8
**Confidence:** 2

**Review:**

TL;DR. a generalization of the mixup algorithm to any layer, improving generalization abilities.

* Summary

The manuscript generalizes the mixup algorithm (Zhang et al., 2017) which proposed to interpolate between inputs to yield better generalization. The present manuscript addresses a fairly more general setting as the mixup may occur at *any* layer of the network, not just the input layer. Once a layer is chosen, mixup occurs with a random proportion $\lambda\in (0,1)$ (sampled from a $\mathrm{Beta}(\alpha,\alpha)$ distribution).

A salient asset of the manuscript is that it avoids a pitfall of the original mixup algorithm: interpolating between inputs may result in underfitting (if inputs are far from each others: the interpolation may overlap with existing inputs). Interpolating deep layers of the networks makes it less prone to this phenomenon.

A sufficient condition for Manifold Mixup to avoid this underfitting phenomenon is that the dimension of the hidden layer exceeds the number of classes.

I found no flaw in the (two) proofs. Literature is well acknowledged. In my opinion, a clear accept.

* Major remarks

- There is little discussion in the manuscript about which layers should be eligible to mixup and how such layers get picked up by the algorithm. I would suggest elaborating on this.
- References: several preprints cited in the manuscript are in fact long-published. I strongly feel proper credit should be given to authors by replacing outdated preprints with correct citations.
- I find the manifold mixup idea to be closely related to several lines of work for generalization abilities in machine learning (not just for deep neural networks). In particular, I would like to read the authors' opinion on possible connection to the vicinal risk minimization (VRM) framework, in which training data is perturbed before learning, to improve generalization (see, among other references, Chapelle et al., 2000). I feel it would help improve supporting the case of the manuscript and reach a broader community.

* Minor issues

- Tables 1 and 3: no confidence interval / standard deviation provided, diminishing the usefulness of those tables.
- Footnote, page 4: I would suggest to add a reference to the consistency theorem, to improve readability.

---

> ### Author Response · Authors · 2018-11-11
> **Thanks for your feedback**
>
> “- There is little discussion in the manuscript about which layers should be eligible to mixup and how such layers get picked up by the algorithm. I would suggest elaborating on this.”
>
> We performed a new experiment to directly study this.  Because the theory in section 3 assumes that the part of the network after mixing is a universal approximator, there is a sensible case to be made for not mixing in the very last layer.
>
> For this experiment, we evaluated PreActResNet18 models on CIFAR-10 and considered mixing in a subset of the layers, we ran for fewer epochs than in the paper (making the accuracies slightly lower across the board), and we decided to fix the alpha to 2.0 as we did in the paper for manifold mixup.  We considered different subsets of layers to mix in, with 0 referring to the input, 1/2/3 referring to the output of the 1st/2nd/3rd resblocks respectively.  For example {0,2} refers to mixing in the input layer and the output of the 2nd resblock.  {} refers to no mixing.
>
> Layers: Test Accuracy
> {0,1,2}:    96.73%
> {0,1}:       96.40%
> {0,1,2,3}: 96.23%
> {1,2}:       96.14%
> {0}:          95.83%
> {1,2,3}:    95.66%
> {1}:          95.59%
> {2,3}:       94.63%
> {2}:          94.31%
> {3}:          93.96%
> {}:            93.21%
>
> Essentially, it helps to mix in more layers, except for the later layers which hurts to some extent - which we believe is consistent with our theory.
>
> “- References: several preprints cited in the manuscript are in fact long-published. I strongly feel proper credit should be given to authors by replacing outdated preprints with correct citations.”
>
> We’ve updated all of the references to the conference/journal citations.  See the new version of the paper uploaded.  In the future it would be nice if arXiv could also list the bibtex for a conference/journal version, because these are often not easy to look up (for example, for older ICLR conferences it was hard to find the bibtex).  Google scholar does not help because it often only lists the first instance of the paper, which is usually arXiv.
>
> “I find the manifold mixup idea to be closely related to several lines of work for generalization abilities in machine learning (not just for deep neural networks). In particular, I would like to read the authors' opinion on possible connection to the vicinal risk minimization (VRM) framework, in which training data is perturbed before learning, to improve generalization (see, among other references, Chapelle et al., 2000). I feel it would help improve supporting the case of the manuscript and reach a broader community.“
>
> The fundamental question of interest to us here is how deep networks behave when evaluated on points which are off of the data manifold.  Vicinal risk minimization (Chapelle 2000), which you refer to, definitely seems like an improvement over ERM, but it seems like it’s very dependent on our ability to select the right “vicinity”.
>
> Our intuition is that our models should still be able to classify well off of the data manifold (just meaning points x where p_data(x)=0), by identifying factors and structural elements that are shared with the training distribution.  VRM can deal with this if the vicinity covers points which are off of the manifold but doesn’t include points which change the class identity.  In practice selecting this can be quite difficult.  Defining the vicinity as a spherical-Gaussian around the data points is unlikely to capture much of the space that exists off of the data manifold (or at least, reach these points with reasonable probability) while avoiding class overlap.
>
> The “AutoAugment” paper (Cubuk 2018) proposed to learn such augmentations with a neural architecture search procedure (i.e. manually training submodels with different augmentation schemes and selecting those which lead to better generalization), although this is quite expensive and may be difficult to scale beyond a sequence of fixed augmentations.

---

> ### Author Response · Authors · 2018-11-28
> **Feedback on Rebuttal**
>
> Hello,
>
> Can you give any feedback on our rebuttal?  We've tried to address specific concerns, especially related to the effect of varying the alpha hyperparameter.  Additionally we fixed the preprint issue.
>
> If there is anything else that we could do that would effect your confidence or views on the paper, we'd be happy to take a look at it.

---

### Official Review · AnonReviewer2 · 2018-10-31

**Rating:** 4
**Confidence:** 4

**Review:**

The paper proposes a novel method called Manifold Mixup, which linearly interpolating (with a careful selected mixing ratio) two feature maps in latent space as well as their labels during training, aiming at regularizing deep neural networks for better generalization and robust to adversarial attacks. The authors experimentally show that networks with Manifold Mixup as regularizer can improve accuracy for both supervised and semi-supervised learning, are robust to adversarial attacks, and obtain promising results on Negative Log-Likelihood on held out samples.

The paper is well written and easy to follow. Various experiments are conducted to support the contributions of the paper.  Nevertheless, the technical novelty seems a bit weak to me. The method basically moves the interpolating process from input space as in MixUp to randomly selected hidden states. More importantly, some of the paper’s claims are not very convincing to me in its current form.

Major remarks:

1.	The authors suggest that Mixup can suffer from interpolations intersecting with a real sample, but how Manifold Mixup can avoid this issue is not very clear to me.
The authors theoretically prove that with the proposed training cost in Manifold Mixup, the representation for each class will lie on a subspace of dimension dim (h) –d +1 (h and d are the hidden dimension and number of classes, respectively). I did not get the idea of how such dimension reduction relates to the ‘’flattening’’ of the manifold and in particular how such representations (representations for each class “concentrating into local regions”) can avoid the class collision issues as that in Mixup.
Experimentally, from Figures 3 and 4, it seems the class collision issue could be worse than that of Mixup. For example, for mixing ratio of 0.6 (meaning the created image has almost half labels from the two original images), MixUp clearly shows, for instance in the second row, that there are two overlapped images (Horse and Plane), but Manifold Mixup seems to have only the Plane in the mixed image with a soft label.

2.	The observations of mixing in the hidden space is better than mixing in the input space seem to contradictive to the observations by Mixup, it would be very useful if the paper can make that much clear to the readers. I would suggest that the authors fully compare with MixUp in the supervised learning tasks, namely using all the datasets (including ImageNet) and networks architectures used in MixUp for supervised learning. In this way, the paper would be much more convincing because the proposed method is so close to MixUp and the observation here is contradictive.
3.	I wonder how sensitive is the parameter Alpha in Manifold Mixup. For example, how the mixing rate Alpha impacts the results for NLL and Semi-supervised learning in section 5.2?
4.	It would be useful to also present the results for SVHN for supervised learning since the Cifar10 and Cifar100 datasets are similar, and the authors have already used SVHN for other task in the paper.

Minor remarks:

1.	In Table2, the result from AdaMix seems missed.
2.	Why not using Cifar100, but with a new dataset SVHN for the semi-supervised learning in section 5.2?
3.	In related work, regarding regularizing deep networks by perturbing the hidden states, the proposed method may relate to AgrLearn (Guo et al., Aggregated Learning: A Vector Quantization Approach to Learning with Neural Networks) as well.

---

> ### Author Response · Authors · 2018-11-07
> **Motivation for why Manifold Mixup Works**
>
> Thank you for your review.  We will post a more detailed response with new experimental results soon, but I want to quickly address issues related to the motivation for why manifold mixup works.  We also updated the paper with a new appendix section H (page 20) which discusses this in more detail and gives an illustration.
>
> “Mixup can suffer from interpolations intersecting with a real sample, but how Manifold Mixup can avoid this issue is not very clear to me … The observations of mixing in the hidden space is better than mixing in the input space seem to contradictive to the observations by Mixup, it would be very useful if the paper can make that much clear to the readers”
>
> You are correct that manifold mixup works through a mechanism which is very different from input mixup, which I think is actually what makes it interesting.
>
> With input mixup, if the interpolations between two points of the same class intersect with points from a different class (or interpolations are inconsistent), this leads to underfitting and poor performance.  You can see this in the center column of figure 1.  However with manifold mixup, the hidden states of the network are learned, such that these inconsistent interpolations are avoided.
>
> To illustrate, let’s imagine that you have a binary classification problem with 2 examples from class A and 2 examples from class B.  Let’s suppose that we perform manifold mixup in a single 1-dimensional hidden layer.  Let’s say that the points from A are both at h=0.  Where can the points from B be located for the interpolations to all return the same label?  If the points from class B have different h values, then the interpolations must be inconsistent.  For example if one point from class B is at h=1 and one point from B is at h=2, then the point h=1 will either be labeled as 100% class B or it will be labeled as 50% class B / 50% class A.  This will cause manifold mixup to have error, and the only way for it to avoid this is to learn the hidden states such that all examples from each class maps to the same point.  This is what needs to happen if we have a 1D hidden space and 2 classes.  For higher dimensional hidden spaces, a similar phenomenon occurs but it is much less restrictive.
>
> Section 3 provides exact conditions for these inconsistent interpolations to be completely avoided.  Essentially, the representations for each class need to “flatten” so that they don’t have any variation in directions which point towards other classes (you can imagine that this would lead to inconsistent interpolations because some points of the same class would have different distances to points from the other classes).  Figure 1c/1f shows exactly how this happens in a toy problem.
>
> Moreover in section 5.1 we presented an experiment where we train with manifold mixup, but don’t pass gradient to layers before the layer where we mix (however all layers are still trained, as the layer to mix in is randomly selected on each update) - and this made accuracy much worse.  This is strong evidence that it is important for manifold mixup to learn to change the representations to make interpolations consistent.
>
> Why is it desirable for manifold mixup to change the representations to avoid inconsistent interpolations?  The first reason is that it can help to avoid underfitting, but another reason is that the way to make interpolations consistent is to make the representations for each class more concentrated, which can only be accomplished by forcing the network to learn more discriminative features in earlier layers.
>
> Please let me know if anything is unclear here, if you’re uncertain about part of the argument, or if there is any other type of illustration/figure that would be helpful.

---

> ### Author Response · Authors · 2018-11-13
> **Thanks for your Feedback**
>
> Remarks 1/2 are addressed in the previous comment "Motivation for why Manifold Mixup Works".
>
> Remark 3: “I wonder how sensitive is the parameter Alpha in Manifold Mixup. “
>
> We didn’t tune alpha very carefully, and used alpha=2.0 in all cases except for supervised learning with the large PreResNet152, where we performed better with larger alphas.  Our general experience is that manifold mixup helps over a wide range of alphas but manifold mixup benefits more from larger alphas, especially when using a larger model.
>
> Nonetheless we performed a new experiment for the rebuttal where we trained a PreResNet18 on CIFAR-10 with a range of alphas.
>
> Baseline (no mixing):       93.21%
>
> Manifold Mixup (α=0.5):  96.12%
> Mixup (α=0.5):                   95.75%
>
> Manifold Mixup (α=1.0):  96.10%
> Mixup (α=1.0): 	             95.84%
>
> Manifold Mixup (α=1.2):  96.29%
> Mixup (α=1.2): 	             96.09%
>
> Manifold Mixup (α=1.5):  96.35%
> Mixup (α=1.5):                   96.06%
>
> Manifold Mixup (α=1.8):  96.45%
> Mixup (α=1.8): 	             95.97%
>
> Manifold Mixup (α=2.0):  96.73%
> Mixup (α=2.0): 	             95.83%
>
> Manifold Mixup outperformed Input Mixup for all alphas in the set (0.5, 1.0, 1.2, 1.5, 1.8, 2.0) - indeed the lowest result for Manifold Mixup is better than the worst result with Input Mixup.  Note that Input Mixup’s results deteriorate when using an alpha that is too large, which is not seen with manifold mixup.
>
> Remark 4: “It would be useful to also present the results for SVHN for supervised learning since the Cifar10 and Cifar100 datasets are similar, and the authors have already used SVHN for other task in the paper.”
>
> We ran new experiments on SVHN using the training set without the “extra” data.  We used PreActResNet-18 and used the exact same setup as with CIFAR-10.
>
> Method: Test Accuracy
> Manifold Mixup (α=2.0):  98.10
> Manifold Mixup (α=1.5):  98.08
> Input Mixup (α=1.5):        97.59
> Input Mixup (α=1.0):        97.63
> Input Mixup (α=0.5):        97.74
> Input Mixup (α=0.2):        97.71
> Input Mixup (α=0.05):      97.72
> Input Mixup (α=0.01):      97.70
> Baseline:                             97.78
>
> Minor Remark 2: “Why not using Cifar100, but with a new dataset SVHN for the semi-supervised learning in section 5.2?”
>
> For SSL. cifar10 (with 4k labelled samples) and SVHN (1K labelled samples) have emerged as the standard benchmark datasets and they have been used to compare all of the recent state-of-the-art methods, so we followed the same setup.  We used the standard semi-supervised setup and used the exact same architectures from (Oliver 2018) “Realistic Evaluation of Deep Semi-Supervised Learning Algorithms”, which evaluated on SVHN and CIFAR-10.
>
> Minor Remark 1/3: “In Table2, the result from AdaMix seems missed… AgrLearn missed”
>
> (Note: updated this rebuttal section on 11/21)
> Note that these were released after our method’s preprint was released and they cite our method, so this is why we originally did not have it in our related work.  Nonetheless the our paper has been updated to discuss Adamix and Agrlearn in the related work.  AdaMix reports 3.52% error on CIFAR-10 and 20.97% error on CIFAR-100.  AgrLearn reports 2.45% on CIFAR-10 and 20.21% on CIFAR-100.  We report 2.38% error on CIFAR-10 and 20.39% error on CIFAR-100.  Note that AgrLearn was used together with Input Mixup (Zhang 2018) on CIFAR-10, so their method may also be complementary with Manifold Mixup as well.  This could be an interesting area for future work.
>
> I think how the methods are related is an interesting question. AdaMix only interpolates in the input space, and they report that their method hurt results significantly when they tried to apply it to the hidden layers.  Thus the methods likely work for different reasons and might be complementary.

---

> ### Author Response · Authors · 2018-11-19
> **Thanks for feedback - is there anything additional that we could do?**
>
> Hello,
>
> Thanks again for your feedback.  Our new experiments directly address the empirical questions for #3/#4 (effect of alpha and SVHN).  We also ran a new experiment for Reviewer-1 which studied the effect of the choice of layers to mix in.
>
> For the conceptual issues about manifold mixup (#1/#2), is there any chance that you could give us more details or feedback on them?  This is really important to us, and if anything is in error or not argued convincingly, it would be great to understand better.
>
> Are there any experiments (especially related to the conceptual properties of manifold mixup) that you would be interested in or that would make the arguments more convincing or that would resolve any remaining issues?
>
> Your feedback has already been very helpful in making the paper better (for example, the new appendix H and Figure 10 illustrating how inconsistent interpolations can be avoided) and if you have any more feedback it could be really helpful for us.

---

> > ### Comment · AnonReviewer2 · 2018-11-22
> > **my main concern remains and I am willing to increase my score if you could address it.**
> >
> > Thank you for your feedback on my review. It addressed some of my concerns. Nevertheless, I am still not fully convinced that the proposed method address Mixup’s collision issue as claimed in your paper’s Abstract.
> >
> > I am glad to see that your method is less sensitive to the pre-defined Alpha than Mixup, so I think you may want to further emphasize that in your paper. Also, the results from SVHN are helpful, though I did not see them in the revision. It would be beneficial to have these good results in the paper.
> >
> > However, my main concern of the paper still remains. That is the claim that the synthetic interpolations generated by Manifold Mixup will not collide with a real sample. The new Section H really makes your point much clearer, but below please find my argument.
> >
> > I agree that if all class manifolds can be “flattened” (BTW, it may be a good idea to further clarify or define the meaning of “flatten”), the collision issue can be addressed. However, before reaching the goal of “flatten” manifolds, your method in fact uses mixed synthetic samples for training. That is, at the earlier stages of the training, collided or conflicted samples are used to chase the “flatten manifold” goal. This means that the model is trained with very noisy data, which may contain synthetic, soft-labeled samples which are intersected or collided with other real samples. When training with collided samples, your training loss could be high, which may prevent you from “flattening” the manifolds.  This issue could be worse when coping with data with a large number of classes such as Cifar100 or ImageNet.
> >
> > I wonder if the following suggestions could further help improve the paper.
> > 1.	Plot all the training loss (with synthetic samples) in the supervised cases.
> > 2.	Provide more analysis on the Cifar100 data. This is because I suspect that the collision issue could be worse when handling datasets with a large number of classes. Ideally, it would be very convincing to have results from ImageNet, which has 1000 classes. BTW, this is also a suggestion from AnonReviewer3. I think it is a really good suggestion.
> > 3.	Any thoughts on my comment regarding Figure 4, which seems to be an indication of collision issue? On the other hand, I have to admit that my observation could be a bit subjective in this case.
> >
> > In short, the paper’s main novelty and contribution is addressing the collision issue in Mixup. I am willing to increase my score if you could address my main concern here.

---

> > > ### Author Response · Authors · 2018-11-26
> > > **Addressing Main Concerns (1)**
> > >
> > > “I am glad to see that your method is less sensitive to the pre-defined Alpha than Mixup, so I think you may want to further emphasize that in your paper. Also, the results from SVHN are helpful, though I did not see them in the revision. It would be beneficial to have these good results in the paper.”
> > >
> > > Thanks, we just added SVHN results (Table 6) to the revision as well as the new analytical experiments as you’ve suggested (Appendices J&K).
> > >
> > > “it may be a good idea to further clarify ... the meaning of ‘flatten’”
> > >
> > > When we refer to flattening, we mean that the class-specific representations have reduced variability in some directions.  Our new spectral analysis in Appendix I makes this more concrete and general, in that we ran a singular value decomposition on the class-specific representations, and found that most of the singular values were greatly reduced.  This has a specific geometric interpretation in which the shape of the class-specific representations can be seen as being more like an ellipsoid (where many singular values are smaller) and less spherical.  Thus we can see it as a flattening.  Note also that our SVD analysis in Appendix I confirms that there is only a class-specific flattening, and the overall representation space is not flattened, which is in line with the intuition that variability is removed in directions which point towards the representations of other classes.
> > >
> > > Section 3 characterizes a sufficient condition for inconsistent interpolations to be avoided in which some directions lose variability completely, and thus we can see those directions as “flattened”.
> > >
> > > “However, my main concern ... is the claim that the synthetic interpolations generated by Manifold Mixup will not collide with a real sample ... at the earlier stages of the training, collided or conflicted samples are used to chase the “flatten manifold” goal. ... training loss could be high, which may prevent you from “flattening” the manifolds.”
> > >
> > > Thanks for bringing this point up.  If we understand correctly, your point is that even if hypothetically the points could be arranged to avoid collisions (i.e. section 3), this could be difficult to achieve in practice especially in the early parts of training where performance on the task is poor.
> > >
> > > Let us suppose X1, X3, X3 are examples from three different classes A, B and C respectively. And let us suppose h1, h2 and h3 are the hidden representations of these samples at layer "h" . Now let us suppose we interpolate h1 and h2 such the interpolation collides with the h3. That is h_interpolated = lambda*h1 + (1-lambda)*h2 = h3 .
> > >
> > > Now assuming that we are training the model with these two samples (h_interpolated, lambda*A + (1-lambda)*B) and (h3, C).
> > >
> > > It is indeed a collision for the layers above the layer "h", since they are being fed with the equal h-representation but different outputs. But since, when we do the parameter update, the gradient passes through the entire network, and hence the network below the layer "h" will adapt itself such that both the samples (h_interpolated, lambda*A + (1-lambda)*B) and (h3, C) are satisfied. And this can be done only if  h1, h2 and h3 are changed in such a way that interpolation between h1 and h2 (h_interpolated) does not collide with h3. So supposedly, if in the next update, the interpolation happens between the hidden states of same samples X1 and X2,  it will not collide with the hidden state of sample X3.
> > >
> > > It is worth noting that while section 3 gives sufficient conditions for this to happen perfectly, in practice,  the model should be able to minimize the inconsistent interpolation problem even if it can’t do it perfectly.  For example, the highest loss type of inconsistent interpolation would involve interpolating between two points from class A and that overlapping with a point from class B.  This point would be given labels of 100% A (interpolating) and 100% B (on that real point) which would lead to very high training error.  But if the model could move the B point slightly off the interpolation, it would reduce the loss quite a lot even if the interpolations aren’t perfectly consistent.
> > >
> > > We also conducted a simple new experiment resulting in some new animated gifs, where we treat each hidden state as a learnable parameter (plus a small amount of gaussian noise) and where they are all initialized randomly, such that the two classes are totally entangled initially.  We can see that the gradient easily can learn to pull the two classes apart, even though the initial states overlap completely on the first step.  (note that I think this process will be even easier in a high dimensional space).  This provides some insight into what happens in the scenario that you’re concerned about - where the states are very noisy at the beginning of training:
> > >
> > > https://media.giphy.com/media/XIEcKAfXr73epPIcOR/giphy.gif
> > >
> > > https://media.giphy.com/media/24lp18V63om8G0dvvg/giphy.gif
> > >
> > > https://media.giphy.com/media/1wXeQi6xHO4UKMnG5s/giphy.gif

---

> > > ### Author Response · Authors · 2018-11-26
> > > **Addressing Main Concerns (2)**
> > >
> > > "I wonder if the following suggestions could further help improve the paper ... Plot all the training loss (with synthetic samples) in the supervised cases.”
> > >
> > > We added a new plot to the paper (Figure 9 of Appendix C) showing the training loss curves for mixing in different levels on CIFAR-10 (each experiment used alpha=2.0, model is PreActResNet18).  The results of this are clear and consistent over the course of training.  At the end of the 30th epoch of training, the train losses are:
> > >
> > > {0}: 0.155
> > > {1}: 0.146
> > > {2}: 0.127
> > > {3}: 0.112
> > > {0,1,2,3}: 0.144
> > > {0,1,2,3}, blocking before mixing: 0.164
> > >
> > > Thus we can see that relatively early in training, the lowest training error comes from mixing in the deepest layer and a much higher loss comes from mixing in the input layer.  Intriguingly, an even higher loss is obtained from mixing in a random layer, but blocking the gradient before the mixing layer.
> > >
> > > “2.Provide more analysis on the Cifar100 data. This is because I suspect that the collision issue could be worse when handling datasets with a large number of classes. Ideally, it would be very convincing to have results from ImageNet, which has 1000 classes. BTW, this is also a suggestion from AnonReviewer3. I think it is a really good suggestion….”
> > >
> > > We did two things to address this: first we analyzed the training loss for CIFAR100, showing that mixing in deeper layers greatly reduces training loss (Figure 10 of appendix C)
> > >
> > > Mixing in different layers on CIFAR-100 (train cross-entropy/error at 30 epochs):
> > >
> > > {0}: 0.0357
> > > {1}: 0.0341
> > > {2}: 0.0332
> > > {3}: 0.0276
> > > {0,1,2,3}: 0.0333
> > >
> > > Secondly, we have results on ImageNet showing significant improvement with Manifold Mixup.  Imagenet has some unique challenges.  Perhaps most importantly, distributed training on imagenet typically uses very large batch sizes.  Manifold Mixup samples a lot of variables randomly once per minibatch in our usual formulation (i.e. one lambda sampled per batch and one layer) and in practice we found that on imagenet this led to a lot of variance between the updates and slowed down training.  To address this we sampled a different lambda for each pair of examples in the batch, which made the loss curves much smoother and made convergence similar to Input Mixup.  With all models we used the same hyperparameters except for the choice of layer to mix in, and we used alpha=0.2.  In all cases we used a ResNet50 and trained for 200 epochs.  While our baseline is somewhat weak, these results suggest that Manifold Mixup can still outperform Input Mixup when the number of classes is large.
> > >
> > > Model: Top-1 Validation Accuracy / Top-5 Validation Accuracy
> > > Baseline: 75.462 / 92.628
> > > Input Mixup: 75.944 / 92.844
> > > Manifold Mixup {0,1,2}: 76.102 / 92.870
> > > Manifold Mixup {0,3}: 76.032 / 92.906
> > >
> > > ResNet50 is a relatively small model for Imagenet, and we hope that the performance gain (improvement in test accuracy) with Manifold mixup will be more with larger models, since we achieved significantly larger gains when using larger models and training for longer on CIFAR10 and CIFAR100.
> > >
> > > “In short, the paper’s main novelty and contribution is addressing the collision issue in Mixup. I am willing to increase my score if you could address my main concern here. “
> > >
> > > The primary goal of the paper is to show how mixing in the hidden layers helps to learn better feature representations.  It is true that as a consequence, this can reduce underfitting relative to Mixup (by avoiding the collision issue), but this is only one consequence.  In our view the more interesting and novel consequence is that this causes a flattening of the learned class-specific representations (see section 3 and Appendix I especially) which encourages the features to be more discriminative.

---

> > > > ### Public Comment · (anonymous) · 2018-12-14
> > > > **The baseline of ResNet-50 on ImageNet**
> > > >
> > > > The baseline of ResNet-50 on ImageNet is lower than that current reproduced one, ~ 76.5% top-1. Eg. mixup: Beyond Empirical Risk Minimization. It is OK that the baseline is a bit lower. However, it would be convincible (for me) if the proposed approach could reach or exceed 76.5% + 0.1% (std).

---

> > > ### Author Response · Authors · 2018-11-26
> > > **Regarding the Visualization in Figure 4**
> > >
> > > “3.	Any thoughts on my comment regarding Figure 4, which seems to be an indication of collision issue? On the other hand, I have to admit that my observation could be a bit subjective in this case.”
> > >
> > > I agree with your basic intuition here, that with Manifold Mixup (Figure 4), many of the points which are given 80% probability of class A actually seem to only have features from class A - for example the fox in the bottom right.  However near the middle, it clearly has a mix of semantically meaningful car and fox attributes.  While I agree that this is important, this is not a collision issue because, although the interpolated points look somewhat unrealistic, they do not look like points from classes other than the two classes being interpolated.
> > >
> > > Another thing to keep in mind is how this visualization was created.  The images shown in the figure 4 are the mappings of interpolated hidden space to the input space using a learned decoder network trained to predict real data points from their hidden states (using square loss). As you pointed out in your previous comment “for mixing ratio of 0.6 (meaning the created image has almost half labels from the two original images), MixUp clearly shows, for instance in the second row, that there are two overlapped images (Horse and Plane), but Manifold Mixup seems to have only the Plane in the mixed image with a soft label. ” , it is true that some of the images seem to have attribute from only one class, but will be given soft labels 50% from class A and 50% from class B.  However, this may be a limitation of the decoder network we used and its fidelity. That is, it is possible that the decoder network was not able to map the interpolated hidden space to an image with 10% horse attributes, even though the interpolated hidden space had some attributes from the horse class.

---

> > > ### Comment · AnonReviewer2 · 2018-11-30
> > > **Concerns After Rebuttal**
> > >
> > > I really appreciate the authors’ rebuttal. However, I am not convinced that the proposed Manifold Mixup scheme in its current form can avoid sample collision, for the following reasons.
> > >
> > > My understanding is that in order for the proposed method to work as claimed, the network layers below the layer chosen for mixing samples needs to be powerful enough to drive the training error close to zero. This requirement is consistent with one of the public comment posted by one author of AdaMixup, which first introduces and formally analyzes the concept of manifold collision issue in Mixup. Also, this requirement seems to be further confirmed by the authors’ new observations of “we found that mixing in deeper layers successfully reduces training error (with greater reduction for deeper layers)”.  In other words, “a randomly chosen layer per minibatch to perform mixup” as implemented by the current Manifold Mixup approach, will not be able to avoid sample collision. In addition, a ResNet as used by the paper will not be able to attain this collision avoidance goal; a network with tailored layers (with sufficient modeling capability) below the mixup layer is needed. In this sense, unlike Dropout or Batch Normalization, Manifold Mixup is NOT a plug-and-play regularization scheme.
> > >
> > > The current form of Manifold Mixup seems to have two conflict objectives to me. On one hand, it requires the mixing layer to be closer to the output layer in order to reduce the collision issue to generate informative representations. On the other hand, mixing close to the output layer will have the negative impact on regularization. A further study on this tradeoff would be very beneficial.

---

> > > > ### Author Response · Authors · 2018-12-01
> > > > **Thanks for continuing discussion / feedback (1)**
> > > >
> > > > Thanks for writing back - we really appreciate the feedback as well as the effort that goes into continuing the discussion.  It means a lot to us.  We also think that we now have a pretty clear understanding of what your objection is.
> > > >
> > > > We agree with the technical claims in your response.  However, reducing collision is NOT the main point or contribution of the paper.  We believe that our main contribution is that manifold mixup dramatically changes the representations learned by deep networks: the distribution of the representations of the examples of each class becomes flattened and concentrated, and more of the hidden space corresponds to less confident classifications.  We have direct empirical evidence for this happening, even in the early layers of the network (spectral analysis in appendix I on MNIST).  Note that this is very different from both Mixup and Adamix, as both of those methods still only interpolate in the input space - and our hidden representations look completely different from Input Mixup or other well known regularizers.
> > > >
> > > > Thanks for helping us make this a better paper by tightening the actual claim of the paper, which is not of perfectly preventing inconsistent interpolations (collisions) , but rather of changing the distribution of representations in a way that is useful for classification, making Manifold Mixup a significant piece of research. We will revise the abstract accordingly and move the emphasis onto what we think is more important.  Here is the proposed and more modest new abstract, highlighting  our contributions more clearly and accurately:
> > > >
> > > > “Deep networks often perform well on the data distribution on which they are trained, yet give incorrect (and often very confident) answers when evaluated on points from off of the training distribution. This is exemplified by the adversarial examples phenomenon but can also be seen in terms of model generalization and domain shift. Ideally, a model would assign lower confidence to points unlike those from the training distribution. We propose a regularizer which addresses this issue by training with interpolated hidden states and encouraging the classifier to be less confident at these points. Because the hidden states are learned, this has an important effect of encouraging the hidden states for a class to be concentrated and flattened with more of the volume of the hidden space mapping to lower confidence classifications.
> > > > This concentration of the class-specific representations can be seen as making the features in earlier layers more discriminative. We prove some exact conditions on how Manifold Mixup changes the representations for a sufficiently deep layer: specifically that there is a flattening effect related to the number of classes and the number of hidden units.   We back up this theoretical analysis of the ideal case by conducting an empirical spectral analysis of the learned representations, showing that this flattening occurs even when we mix immediately following the first hidden layer.  We show that despite requiring no significant additional computation, Manifold Mixup achieves large improvements over strong baselines in supervised learning, robustness to single-step adversarial attacks, semi-supervised learning, and Negative Log-Likelihood on held out samples.”
> > > >
> > > > There are a few other points where the writing would need to be changed to be consistent with that, for example the caption of figure 1.  We will also change all claims of “avoiding” or “removing” inconsistent interpolation to the more modest and strongly supported claim that inconsistent interpolations are reduced as part of how the hidden layers representations are changed.

---

> > > > ### Author Response · Authors · 2018-12-01
> > > > **Thanks for Continuing Discussion / Feedback (2)**
> > > >
> > > > "I really appreciate the authors’ rebuttal. However, I am not convinced that the proposed Manifold Mixup scheme in its current form can avoid sample collision, for the following reasons.”
> > > >
> > > > We agree with this, especially because we still do mix in the input layers.
> > > >
> > > > “My understanding is that in order for the proposed method to work as claimed, the network layers below the layer chosen for mixing samples needs to be powerful enough to drive the training error close to zero. ... “a randomly chosen layer per minibatch to perform mixup” as implemented by the current Manifold Mixup approach, will not be able to avoid sample collision.“
> > > >
> > > > Manifold Mixup consistently improves test accuracy over baseline for any combination of layers to mix in (see our response to R1).  And in fact substantial regularization is achieved even when we mix after the 3rd resblock, which is near the end of the network.  At the same time it also helps to mix in earlier layers.
> > > >
> > > > I think that our goal and procedure is very different from adamix.  And for what it’s worth, the ideas seem very complementary to us.  Perhaps one could train with Manifold Mixup, but when mixing in the input layer (and also the 1st hidden layer) one could use AdaMix on those updates.  When mixing in the deeper layers, we would have the desirable effect of Manifold Mixup: learning flatter representations and broader regions in the hidden space with lower confidence classifications - and the use of Adamix would let us use larger alphas when we do mix in the input layer.
> > > >
> > > > Thus we don’t see any conflict between the ideas.  Our paper is about mixing in the hidden layers, and about how the dynamics of mixing in the hidden layers effects what they learn.  This is related to reducing inconsistent interpolations, but it is primarily a “means to an end” of changing how the hidden layers represent the data.
> > > >
> > > > “In addition, a ResNet as used by the paper will not be able to attain this collision avoidance goal; a network with tailored layers (with sufficient modeling capability) below the mixup layer is needed. In this sense, unlike Dropout or Batch Normalization, Manifold Mixup is NOT a plug-and-play regularization scheme.”
> > > >
> > > > We understand your point here, but if you look at it from the perspective of test accuracy on real datasets, it does seem to be "plug and play".  For example in our CIFAR-10 analysis of the mixing layer given in the response to R1 (Appendix K), you can see that every mixing combination improves over baseline, and most configurations improve over input mixup.
> > > >
> > > > “The current form of Manifold Mixup seems to have two conflict objectives to me. On one hand, it requires the mixing layer to be closer to the output layer in order to reduce the collision issue to generate informative representations. On the other hand, mixing close to the output layer will have the negative impact on regularization. A further study on this tradeoff would be very beneficial."
> > > >
> > > > Our strongest theoretical guarantee only holds for deeper layers, but intuitively even for early layers the effect of representations being flattened can still occur, and there’s strong evidence that it still has a positive effect on regularization.  Also our analysis in appendix I shows that this flattening happens empirically even following a single hidden layer.
> > > >
> > > > At a high level, we strongly agree that we need to soften and reduce some of our claims for what Manifold Mixup accomplishes (specifically we don’t want to claim that our method completely avoids collision).  At the same time we still think that our contribution is significant and well supported - which is that Manifold Mixup changes how deep networks represent information by flattening the class-specific representations and assigning low confidence classifications to more of the space.

---

> > > > ### Author Response · Authors · 2018-12-02
> > > > **Feedback on Response**
> > > >
> > > > Hello,
> > > >
> > > > Do you have any additional feedback on our response?  We greatly appreciate the time that you've given to discussing the paper with us so far and providing feedback.

---

> > > > ### Author Response · Authors · 2018-12-02
> > > > **Thanks for Feedback and Reviewing**
> > > >
> > > > We appreciate all of your help in reviewing the paper, and we think that several of the points that you've raised have helped us to make the paper better (such as the new appendix on how the method works and the change of the abstract, though this came after the final deadline for updating the paper).
> > > >
> > > > I believe that the deadline for authors to post comments is closing soon, but if you could take a look at our most recent response, as well as the new experiments for the rebuttal we'd really appreciate it.
> > > >
> > > > Once again, thanks for all of your help.

---

> ### Author Response · Authors · 2018-11-28
> **Feedback on Response**
>
> Hello,
>
> In your most recent response, you mentioned issues related to the flattening of the manifold and the training loss.  Can you give any feedback on our response to these issues?  We really appreciate it.

---

> ### Author Response · Authors · 2018-12-12
> **Feedback on Rebuttal**
>
> Hello,
>
> Thank you for your time in reviewing and we appreciate your time in discussing with us.  I want to summarize so far:
>
> 1.  For your original major concerns #1/#2, we added a new appendix explaining how Manifold Mixup changes the learned representations intuitively as well as a new spectral analysis of the representations which shows how this change happens empirically, even when mixing follows the first hidden layer.
>
> We also acknowledge that we don't want to overclaim what manifold mixup does here as it still uses interpolations in the input layer and early hidden layers (which could but would not necessarily lead to underfitting on some datasets).  However our results show that it does help empirically to use it in almost any combination of layers, and we have strong evidence that this largely results from manifold mixup changing the learned representations.  We revised the abstract to make this clear.
>
> 2.  We performed new experiments which exactly address major concerns #3 and #4, namely sensitivity to alpha and results on SVHN.
>
> 3.  We addressed the "minor remarks" by adding a more thorough discussion of Adamix and adding results from that paper into our tables.  We have also made a case for why it is distinct from and likely to be complementary to manifold mixup.
>
> While I agree that there is more that could be learned about the method, we have made a substantial effort to address all of your major concerns.  At least half of your major concerns (#3/#4), those related to new experiments, have been perfectly addressed.
>
> Is there anything else that we could do that would address any remaining concerns that you have?  This is very important to us and we really appreciate your feedback so far.

---

### Official Review · AnonReviewer3 · 2018-11-03
**The paper is well written and its tone is notably scientific, though the novelty is limited**

**Rating:** 6
**Confidence:** 4

**Review:**

The tone of the paper is notably scientific, as the authors clearly state the assumptions and all observations, whether positive or negative. That said, the approach itself can be seen as a direct extension of the earlier advanced 'mixup' scheme. In addition to performing data augmentation solely in the input space, their method proposes to train the networks on the convex combinations of the hidden state representations by learning to map them to the convex combinations of their one-hot ground truth encodings.

The results are competitive, in most cases exceeding the current state-of-art. However, the scheme has only been tested on low-res datasets such as MNIST, CIFAR and SVHN while the predecessor (plain 'mixup') also demonstrated improvement over the much larger and high-res ImageNet dataset.

Although their work is not extremely novel, the experiments and observations could serve as a useful extension to this line of research.

Suggestions:
1.  The results on ImageNet would be a useful add-on to really drive home the benefit of their method when we talk of real-world large-scale datasets.
2. The associated functions represented by 'f',  'g' and 'h' change meaning between sec. 2 and sec. 3. It would be more smooth if some consistency in notations was maintained.

---

> ### Author Response · Authors · 2018-11-13
> **Thanks for your feedback**
>
> R3:
>
> “Although their work is not extremely novel, the experiments and observations could serve as a useful extension to this line of research. “
>
> Although novelty is subjective, there is a case that the work is actually quite novel:
>
>    1) We present a novel analysis of how manifold mixup changes representations (section 3) which is totally different from the motivation of mixup (and indeed deals with a completely different problem, as the inputs in input mixup are fixed and cannot be changed by training).
>
>   2) The way that the representations are changed by manifold mixup is to our knowledge fairly unique, not just relative to mixup, but compared to other regularizers as well.  For example if you look at Figure 1 and Figure 6 in appendix B, you’ll see that the way the representations are changed by manifold mixup is not accomplished by four common regularizers: weight decay, batch normalization, dropout, and adding noise to the hidden states.  The representations look completely different, even though all of the methods succeed (to some extent) as regularizers.  More concretely, manifold mixup has the fairly unique effect of concentrating the hidden states of the points from each class and encouraging the hidden state to have broad areas of low confidence between those regions.  This is not accomplished to any appreciable degree by the other regularizers.  This is some evidence that the method by which manifold mixup achieves regularization is fairly unique and worthy of further study.
>
> “The associated functions represented by 'f',  'g' and 'h' change meaning between sec. 2 and sec. 3. It would be more smooth if some consistency in notations was maintained.”
>
> Thanks, that’s a good catch.  Our intent was for g to refer to the earlier part of the network and for f to refer to the later part of the network.  We’ve fixed the notation and uploaded an updated version of the paper.

---

> ### Author Response · Authors · 2018-11-28
> **Feedback on Rebuttal**
>
> Hello,
>
> Can you give any feedback on our rebuttal?

---

> ### Author Response · Authors · 2018-12-02
> **Thanks for feedback**
>
> Hello,
>
> Thank you for taking the time to review our paper.  Through the course of the rebuttal, we have conducted some new experiments which address some of the points that you've raised, and we have also produced some arguments in favor of the novelty of the work.  I think that the spectral analysis of the representations (Appendix I) is especially significant as it shows a significant flattening effect from the use of Manifold Mixup, and no consistent effect from Input Mixup, which is strong evidence in favor of Manifold Mixup working through a novel mechanism.

---

### Public Comment · (anonymous) · 2018-11-11
**Table 3. FGSM replaced by MIM or PGD?**

Do the numbers hold up when you replace FGSM by MIM or PGD perhaps?

---

> ### Author Response · Authors · 2018-11-11
> **PGD**
>
> Hello,
>
> Manifold Mixup improves robustness to the weak FGSM attack but does not provide any robustness to the PGD attack (and I'm guessing for any strong attack).  The same is true for mixup.  This is actually already mentioned in the text at the end of section 5.3 and there is some discussion there on the intuition for this.
>
> Our only goal in including these results is to show that at least in some directions, Manifold Mixup does a better job than Input Mixup at moving the decision boundary away from the data - and not to claim robustness (which would require the decision boundary to move further away in *all* directions).
>
> I think that at least one reason why it is not adversarially robust, is that we only consider interpolations between pairs of points, and thus I don't think there's a reason to believe that these points would cover all of the directions that an adversarial perturbation could take around a data point.

---

> > ### Public Comment · (anonymous) · 2018-11-11
> > **Thanks for the response!**
> >
> > That clears up things! Nice paper!

---

### Public Comment · ~Yongyi_Mao1 · 2018-11-22
**Interesting and Promising**

I am an author of AdaMixup, I am going to post this review with my name revealed :)

I am glad that you cited our AdaMixUp paper (to appear in AAAI 2019). In fact, your work and ours deal with the same problem, the problem we call "manifold intrusion" (where by manifold, we mean data manifold, different from what you mean here.)  The problem is that when using the conventional mixup as a regularization scheme, there is no guarantee that training using mixed samples doest not conflict with training using the original samples. Such a conflict, when arising,  results in under-fitting.  Our approach, AdaMixUp deals with this problem in the input space, and you deal with it in a latent-representation space.

Basically, in my understanding, your approach is to force,  hopefully, the network ```to "untangle" the data in the latent space so that interpolated samples in the latent space does not collide with the original samples in their training objectives. Overall I think this is an interesting idea and a promising direction,  but I do have a few questions, some perhaps more essential than others.

1) In order for your scheme to work as desired, it is required that the network has sufficient capacity when transforming the data into a given latent space/layer. (You theorem only holds under such an infinite capacity assumption).  This implies either a) the appropriate choice of representation layer (on which you apply mixUp) is somewhere near the output, or b) the initial layers of the network are sufficiently complex (e.g. wide). When the overall network isn't very deep and not too wide (but still overfits), there is no guarantee that your scheme will work as an effective regularization scheme. That is, there is no guarantee that the network architecture up to the latent representation layer is capable fitting both the mixUp objective and the regular training objective. In other words, I suppose that your scheme may not be compatible with certain network architectures.  This seems to be a disadvantage of your approach comparing to AdaMixUp.

2） Continuing from above, in practice we use finite-capacity networks. Then it is not clear to what extent the manifold intrusion (or under-fitting)  is resolved with your approach. One measure that can reveal the answer to this is the loss associated with the mixUp samples. If the mixUp loss can be driven to sufficiently low, it says that you more or less have succeeded in avoiding manifold intrusion. But I do not seem to see you show the curve of this loss.

3) I actually do not see the value of pick a random layer to do mixUp.  It seems to me that as long as at one layer the network is capable of transforming the data into the "flattened" manifold (using your word, but actually I think it is not the correct word), that is sufficient. Is there any principle underlying such a randomized strategy?

4) I am not too convinced that your approach outperforms AdaMixUp. (I may look protective of our own, but who would easily accept defeat ? :) Did you compare your scheme with AdaMixUp on the same network structure (say, on CIFAR 100)? And the resNet result you present to compare with ours does not use a network having the same number of layers with ours.

5) I believe that our adaMixUp paper is the first work that pinpointing the manifold intrusion/underfitting problem of MixUp. I think you should give us the deserving credits :)

---

> ### Author Response · Authors · 2018-11-24
> **Thanks for your valuable feedback (1)**
>
> "I am glad that you cited our AdaMixUp paper (to appear in AAAI 2019). ... Our approach, AdaMixUp deals with this problem in the input space, and you deal with it in a latent-representation space. "
>
> I think our main goal is to show that by trying to avoid intrusion, Manifold Mixup learns better hidden representations.  The key thing is the way that the hidden representations are themselves changed.  We demonstrate that this change consists of a flattening of the class-specific representations in theory (section 3), empirically (through the spectral analysis in Appendix I), and on toy datasets where the states can be visualized (Figure 1).  Are these flattened representations better?  One strength is that they encourage the features in earlier layers to be more discriminative and another is that they make the features from real data points more concentrated, which means that more of the hidden space can be assigned lower confidence.
>
> "1) In order for your scheme to work as desired, it is required that the network has sufficient capacity when transforming the data into a given latent space/layer. "
>
> So section 3 gives sufficient conditions for Manifold Mixup to attain zero loss, but we have experimental evidence that this flattening occurs even when these conditions aren't satisfied exactly (especially the spectral analysis in appendix I).  Intuitively this is what we'd expect, because even if the number of hidden dimensions is too small, the different classes could be placed on the surface of a regular polygon, so that interpolations at least can avoid intersecting with real data points (you can actually see this happening in Figure 7 where we used 2 hidden units and 5 classes).
>
> At the same time, there is another motivation to Manifold Mixup unrelated to intrusion, which is that the higher level hidden layers will generally learn more semantically meaningful features, so interpolating that space will produce more meaningful mixes (and perhaps more similar to the points that can occur in the test set).
>
> I think there is significant strength in AdaMix's approach of still mixing in the input space but avoiding intrusions by learning where not to mix.  One significant strength is that intrusion can be avoided while mixing still occurs entirely in the input space - which may provide stronger regularization than Manifold Mixup in some cases.  For example, I wouldn't be surprised if AdaMix helped a lot with adversarial robustness (as robustness is usually defined in terms of the input space).
>
> "2） Continuing from above, in practice we use finite-capacity networks. ... One measure that can reveal the answer to this is the loss associated with the mixUp samples. If the mixUp loss can be driven to sufficiently low, it says that you more or less have succeeded in avoiding manifold intrusion. But I do not seem to see you show the curve of this loss. "
>
> We added a new plot to the paper (Figure 9) showing the training loss curves for mixing in different levels on CIFAR-10 (each experiment used alpha=2.0).  The results of this are clear and consistent over the course of training. At the end of the 30th epoch of training, the train losses are:
>
> {0}: 0.155
> {1}: 0.146
> {2}: 0.127
> {3}: 0.112
> {0,1,2,3}: 0.144
> {0,1,2,3}, blocking before mixing: 0.164
>
> Thus we can see that relatively early in training, the lowest training error comes from mixing in the deepest layer and a much higher loss comes from mixing in the input layer.  Intriguingly, an even higher loss is obtained from mixing in a random layer, but blocking the gradient before the mixing layer.

---

> ### Author Response · Authors · 2018-11-24
> **Thanks for your valuable feedback (2)**
>
>
> "3) I actually do not see the value of pick a random layer to do mixUp.  It seems to me that as long as at one layer the network is capable of transforming the data into the "flattened" manifold (using your word, but actually I think it is not the correct word), that is sufficient. Is there any principle underlying such a randomized strategy? "
>
> On the 2d spiral dataset, it worked well to only mix in a single hidden layer, but in general it's difficult to know which layer to mix in.  Mixing later will do a better job of avoiding the intrusion / inconsistent interpolation problem, but may have a more limited effect in terms of regularization on some datasets (which is confirmed experimentally in our response to Reviewer #1 where mixing in multiple layers helped).
>
> ""flattened" manifold (using your word, but actually I think it is not the correct word)"
>
> What we mean by this is that variability is removed in some directions (thus these directions become more "flat"), and this flattening only occurs in the class-specific representations, and the intuition is that variability is reduced in directions which point towards examples from other classes.  The spectral analysis in appendix I provides support for this empirically and the theory in section 3 characterizes this flattening.
>
> "4) I am not too convinced that your approach outperforms AdaMixUp. (I may look protective of our own, but who would easily accept defeat ? :) Did you compare your scheme with AdaMixUp on the same network structure (say, on CIFAR 100)? And the resNet result you present to compare with ours does not use a network having the same number of layers with ours."
>
> We have updated the paper, particularly Table 1 to be more clear on this.  Using the same architecture (ResNet18), Manifold Mixup is better on CIFAR-10 but slightly worse on CIFAR-100 on the same architecture but performs better when using a somewhat deeper network (ResNet34).
>
> One thing is that I don't see why the methods couldn't be used together.  You could use Adamix when you mix in the input layer and not use it when you mix in the later layers (as in manifold mixup).  Perhaps for the 1st or 2nd layer you could use a weakened version of Adamix.
>
> I think the key thing is that both methods improve end-results, but work through very different mechanisms, and have overlapping but distinct goals and priorities.  I think this is how research ought to be.  When you have two methods like "dropout" and "batch normalization", both do act as regularizers and can improve test accuracy, but their mechanisms and motivations are different, and it's important for the community to understand these mechanisms so that they can know where and how to apply them and what to work on in the future.
>
> "5) I believe that our adaMixUp paper is the first work that pinpointing the manifold intrusion/underfitting problem of MixUp. I think you should give us the deserving credits :)"
>
> We agree that it is very important to give proper credit here.  While our preprint was released earlier, it has seen significant revisions after the release of the AdaMix paper and these have more heavily emphasized the concept of intrusion which was formally introduced and received a thorough treatment in the AdaMix paper.

---

### Author Response · Authors · 2018-11-22
**Spectral Analysis of the Learned Representations**

Hello,

Several reviewers have discussed the nature of the "flattening" of representations accomplished by manifold mixup.  Previously our paper had theoretical results (section 3) and visualizations on toy problems with a 2-dimensional hidden layer.  To improve on this, we conducted a new analysis based on the singular value decomposition (SVD) of the representations in a hidden layer.  The goal here is to produce a precise empirical characterization of the flattening effect of Manifold Mixup.  This is added in our new Appendix I (Figures 11-14).

We trained fully-connected models with a bottleneck hidden state (of either 12 or 30 dimensions) on the MNIST dataset.  We considered placing this bottleneck state after 3 hidden layers and after a single hidden layer.  We then performed SVD on those hidden representations to recover the singular values, which we plotted.  We found the effect to be quite strong in both cases: Manifold Mixup reduces the value of the smaller singular values in the class-specific representations.  This suggests that many directions have been 'flattened', and these directions with variance have been removed.  At the same time, when we look at the singular values of the set of all representations (not class specific) we see no clear difference between Manifold Mixup and the Baseline - which is in accordance with the intuition of the proof in section 3 - that variability which is removed is the variability which points in the direction of other classes.  In general the effect of Input Mixup on the singular values was inconsistent, which provides even more evidence that Manifold Mixup operates by a very different mechanism.  Both Manifold Mixup and Input Mixup reduce the size of the largest singular value (spectral norm).

Finally, when placed after a single hidden layer (which we would expect to be a somewhat weak model, and not able to solve the task completely), we still observed a clear flattening effect from Manifold Mixup (Figure 13) but less than when mixing is done in later layers.

When we have referred to flattening, we have meant that the number of dimensions with variability is reduced, and the theory in section 3 gives some conditions for this to happen.  At the same time, this new analysis gives us another way of thinking about flattening in terms of the geometric interpretation of singular value decomposition.  You can think of the U and V matrices as rotations and the singular values (sigma) as a rescaling along dimensions.  Thus reducing some of these singular values can be seen as a flattening effect in those directions.

---

### Author Response · Authors · 2018-11-26
**Rebuttal Summary and Highlights**

We thank the reviewers for their feedback, and we believe that it’s done a great deal to help us to make the paper better.  We want to provide a summary of the new results that we’ve produced and how they relate to reviewer feedback.

1. Novelty: In our opinion, Manifold Mixup works through a mechanism which is very different from Input Mixup, and we have analyzed this both theoretically (section 3) and empirically (Spectral Analysis in Appendix I).  Note that this analysis is completely different from the analysis in Mixup, and is indeed almost totally unrelated to that work as it strongly relies on the states being learned.  The way that Manifold Mixup changes the representations is very different from other state of the art regularizers including dropout, batch normalization, injecting noise, input mixup, and weight decay (as shown in Figure 1 and Figure 6).

2.  Flattening of the class conditioned manifolds:  Many reviewers were unsure about what we meant by “flattening” or were skeptical about whether such an effect could occur even when the exact conditions of the theorem in section 3 didn’t hold.  We have clarified that flattening in the class-specific representations refers to a reduction in the within class variability in some directions.  We have added a new Spectral Analysis of the learned representations (Appendix I) which shows that Manifold Mixup significantly reduces many of the class-specific singular values, which essentially results in “flattening” of class-conditional manifolds (we found no consistent effect with Input Mixup).  Importantly we showed that this happens even when we do this analysis immediately following the first hidden layer.  Intriguingly, we also found that the non-class specific representations (i.e. all representations grouped together) shows no flattening with Manifold Mixup, which is consistent with the intuition of the proof in section 3: that variability is removed in directions which point towards the representations of other classes.

3. Inconsistent interpolations or the collision issue in interpolations:  Reviewers were unsure about whether Manifold Mixup could successfully reduce “inconsistent interpolations” (collision issue, where the interpolation between two samples collide with a sample from other class), especially when the number of classes is large.  We ran additional experiments to address this. On CIFAR-100, we found that mixing in deeper layers successfully reduces training error (with greater reduction for deeper layers), Appendix C, Figure 10 (and the same on CIFAR-10, in Figure 9).  Additionally we were able to improve over Mixup on the Imagenet dataset, which has 1000 classes.

4.  Intuitive Explanation of how Manifold Mixup changes the learned hidden states to reduce inconsistent interpolations:  We have added a new Appendix H illustrating a toy example of how the hidden states can be reorganized via Manifold Mixup to avoid the inconsistent interpolations (collision issue).  Additionally in our response for R2, we conducted a simple experiment where we treat the hidden states as independent learned parameters (initialized randomly) to show that gradient descent can separate the classes even when initialized so that they completely overlap initially:

https://media.giphy.com/media/XIEcKAfXr73epPIcOR/giphy.gif

https://media.giphy.com/media/1wXeQi6xHO4UKMnG5s/giphy.gif

https://media.giphy.com/media/24lp18V63om8G0dvvg/giphy.gif


5 . Analysis of Hyperparameters “alpha” and the layers in which mixing is done: As suggested by the reviewers (especially R1 and R2), we conducted additional experiments to address these questions.  In appendix J, we show that Manifold Mixup improves over Input Mixup works for a wide range of alpha values. Furthermore, in appendix K, we show that mixing in multiple layers improves test accuracy.

---

### Public Comment · (anonymous) · 2018-12-01
**A Related Paper by Wang et al.**

A similar idea of interpolation has appeared in an early paper by Wany et al. https://arxiv.org/pdf/1802.00168.pdf. The authors should mention related work.

---

> ### Author Response · Authors · 2018-12-01
> **The link you sent is not working**
>
> Hello,
>
> Thanks for your comment. Can you check the link and send again?

---

> > ### Public Comment · (anonymous) · 2018-12-02
> > **See this one**
> >
> > https://arxiv.org/pdf/1802.00168.pdf

---

> > > ### Author Response · Authors · 2018-12-02
> > > **Thanks**
> > >
> > > Hello,
> > >
> > > We definitely agree that we should have cited this and we sincerely apologize for not doing so.  We will do so immediately when we get a chance to update the paper.  The (Wang et. al 2018) method is interesting and shares some important similarities and differences with Manifold Mixup.  I'll discuss them below for the benefit of readers:
> > >
> > > -A really big difference is that Manifold Mixup interpolates in multiple layers, and in practice we never did it directly at the output layer.  On the other hand the WNLL method (Wang et. al 2018) exclusively operates in the output layer.
> > >
> > > -If I understand it correctly, WNLL frames their algorithm with an Euler-Lagrange equation (Equation 2) where the cost is based on unlabeled data (and the pseuolabels at those points) and the labeled data provide constraints.  In some ways this is related to how we do semi-supervised learning in Manifold Mixup by using a pseudolabel loss on interpolated points between unlabeled data and a normal supervised loss on the labeled data.  However in Manifold Mixup we just add these two losses together and use a weighting between them (closely following other work with consistency losses like VAT (Miyato 2017)).
> > >
> > > -Manifold Mixup uses simple backpropagation through the interpolation, whereas WNLL has a more complicated training procedure.  Critically, WNLL avoids backpropagating through the interpolation procedure, whereas Manifold Mixup presents evidence that backpropagating through the interpolation is essential to why the method works.  Additionally the two methods, as far as I can tell, have rather distinct motivations and especially distinct theory.
> > >
> > > -The regularization effect (when comparing same architecture) is larger when using Manifold Mixup when we compare the same architectures and using the full dataset (for example on CIFAR-10/PreActResNet18 Manifold Mixup is 2.89% error and WNLL is 4.74%, with baseline of 6.21%).  However, it is still quite impressive that WNLL (Wang et. al 2018) achieves such significant gains while only operating in the output layer.  Also Manifold Mixup doesn't have the same experiments with very small amounts of labeled data, and WNLL shows strong results in these cases.

---

### Meta-Review · Area_Chair1 · 2018-12-17
**Needs improvement.**

**Confidence:** 4
**Recommendation:** Reject

**Metareview:**

The paper contains useful information and shows relative improvements compared to mixup. However, some of the main claims are not substantiated enough to be fully convincing. For example, the claims that manifold mixup can prevent can manifold collision issue where the interpolation between two samples collides with a sample from other class is incorrect. The authors are encouraged to incorporate remarks of the reviewers.